



# Regional centroid MT inversion of small to moderate earthquakes in the Alps using the dense AlpArray seismic network: challenges and seismotectonic insights

Gesa Maria Petersen[1,2], Simone Cesca[1], Sebastian Heimann[1], Peter Niemz[1,2], Torsten Dahm[1,2], Daniela Kühn[1,3], Jörn Kummerow[4], Thomas Plenefisch[5], and AlpArray Working Group[+]

[1]GeoForschungsZentrum Potsdam, Potsdam, Germany
[2]University of Potsdam, Institute of Geosciences, Potsdam, Germany
[3]NORSAR, Applied Seismology, Kjeller, Norway
[4]Freie Universität Berlin, Berlin, Germany
[5]Federal Institute for Geosciences and Natural Resources (BGR), Hanover, Germany
[+]For further information regarding the team, please visit the link which appears at the end of the paper.

**Correspondence:** Gesa M. Petersen (gesap@gfz-potsdam.de)

**Abstract.** The Alpine mountains in central Europe are characterized by a heterogeneous crust accumulating different tectonic units and blocks in close proximity to sedimentary foreland basins. Centroid moment tensor inversion provides insight into the faulting mechanisms of earthquakes and related tectonic processes, but is significantly aggravated in such an environment. Thanks to the dense AlpArray seismic network and our flexible bootstrap-based inversion tool *Grond* we are able to test differ-
ent set-ups with respect to the uncertainties of the obtained moment tensors and centroid locations. We evaluate the influence of frequency bands, azimuthal gaps, input data types and distance ranges and study the occurrence and reliability of non-DC components. We infer that for most earthquakes (Mw≥3.3) a combination of time domain full waveforms and frequency domain amplitude spectra in a frequency band of 0.02-0.07 Hz is suitable. Relying on the results of our methodological tests, we perform deviatoric MT inversions for events with Mw>3.0. We present here 75 solutions and analyse our results in the
seismo-tectonic context of historic earthquakes, seismic activity of the last three decades and GNSS deformation data. We study regions of high seismic activity, namely the western Alps, the region around Lake Garda, the SE Alps, besides clusters further from the study region, in the northern Dinarides and the Apennines. Seismicity is particularly low in the eastern Alps and in parts of the central Alps. We apply a clustering algorithm to focal mechanisms, considering additional focal mechanisms from existing catalogs. Related to the NS compressional regime, E-W to ENE-WSW striking thrust faulting is mainly observed
in the Friuli area in the SE Alps. Strike-slip faulting with a similarly oriented pressure axis is observed along the northern margin of the central Alps and in the northern Dinarides. NW-SE striking normal faulting is observed in the NW Alps showing a similar strike direction as normal faulting earthquakes in the Apennines. Both, our centroid depths as well as hypocentral depths in existing catalogs indicate that Alpine seismicity is predominantly very shallow; about 80% of the studied events have depths shallower than 10 km.





# 1 Introduction

The Alpine mountains and surrounding areas are known for their complex tectonic setting with a highly heterogeneous lithospheric structure (e.g. Handy et al. (2010, 2015); Schmid et al. (2004); Hetényi et al. (2018)). The mountain range was tectonically shaped by the interaction of Adriatic and European microplates in several stages of compression between the two large converging plates, Europe and Africa (e.g. Schmid et al. (2004, 2008); Handy et al. (2010); Hetényi et al. (2018)). Geological

studies show that the Adriatic plate was upthrusted in the western and central Alps (e.g., Schmid et al. (2008); Handy et al. (2015)). The terranes of the Mesozoic Tethys ocean between Europe and Africa were compressed, rotated, faulted and stacked during the compression stages of the Alpine orogenesis (e.g. Handy et al. (2010)). Along a distance of approximately 700 km between NW Italy and Slovenia, the northern Alps and the southern Alps are separated by the Periadtriatic line or fault system

(e.g. Handy et al. (2005)). Reversals in subduction polarities have been proposed at the transition to both, the Apennines and the Dinarides, while the geometry and orientation of the slab is still controversial (e.g. Hetényi et al. (2018); Handy et al. (2010, 2015); Mitterbauer et al. (2011); Schmid et al. (2004)). Velocity anomalies in the crust and upper mantle reflect this complex crustal structure and geodynamic setting (e.g. Diehl et al. (2009); Fry et al. (2010); Molinari et al. (2015); Kästle et al. (2018); Lu et al. (2020); Qorbani et al. (2020)).

Seismic activity across the Alps is typically characterized by low to moderate magnitude earthquakes. However, large damaging earthquakes have occurred in the past, such as the 1356 Basel earthquake (Meyer et al., 1994) or the 1976 Friuli earthquake (Mw 6.45, Poli and Zanferrari (2018)). Recent seismic activity in the SE Alps is caused by the N-S convergence (2-3 mm/yr) between the Adriatic Plate and Eurasia, which is accommodated by the ENE trending, SSE verging thrust front of the eastern Alps and by NW-SE trending right-lateral strike-slip fault systems in western Slovenia (Poli and Zanferrari, 2018). The wider

Alpine region including parts of the Dinarides and the Apennines stretches across Switzerland, Austria, Liechtenstein, France, Italy, Germany, Slovenia, and Croatia. Most of these countries have national earthquake observatories, research institutes or universities that routinely monitor the regional seismicity. The Swiss Seismological Service (SED) and the Slovenian Environment Agency (ARSO) provide annual reports containing mainly first motion based focal mechanisms (e.g. Diehl et al. (2018) and Ministrstvo za okolje in prostor Agencija RS za okolje (2020)) while for example INGV (Italy), GEOFON (Germany),

EM-RCMT (European-Mediterranean Regional Centroid-Moment Tensors, Pondrelli (2002)), SISMOAZUR (France) as well as gCMT (Lamont-Doherty Earth Observatory of Columbia University, USA) provide moment tensor (MT) solutions in online bulletins for magnitudes above 3.5 or larger (see also data and code availability).

The region can be characterised by compartments with varying tectonic movement in close proximity, as described by many studies of local seismic activity. Focal mechanism in the SW Alps indicate predominantly N-S to NNW-SSE striking normal

faulting (e.g. Nicolas et al. (1998) and Sue et al. (2000)), while in the W Alps strike-slip earthquakes have been observed and explained as a consequence of regional NW-SE compression and NE-SW extension (Maurer et al., 1997). A rotation of the dominant tensional axis over the bending western Alps was recently described by Mathey et al. (2020). The Albstadt Shear



Zone in the NW foreland of the Alps is dominated by NNE-SSW striking sinistral strike-slip events (Mader et al., 2020). In the central Alps, Marschall et al. (2013) observe strike-slip faulting in central Switzerland. NW-SE striking normal faulting is

reported for SE Switzerland (e.g. Marschall et al. (2013); Diehl et al. (2018)). Reiter et al. (2018) provide focal mechanism solutions from P and S polarities and amplitude ratios for the northern central to eastern Alps. They report strike-slip mechanisms and oblique strike-slip mechanisms in the Brenner-Inntal transfer zone, and normal faulting is seen with a strike direction parallel to the Giudicarie fault system. Within this fault system, EW to NE-SW striking thrust faulting with strike-slip components were described by Viganò et al. (2008). The Italian CMT data set provides extensive mechanisms for N Italy. Within the Lake

Garda region and in the SE Alps close to Friuli thrust faulting with ENE-WSW to ESE-WNW strike direction is dominant (Pondrelli et al., 2006; Anselmi et al., 2011; Bressan et al., 1998). East of Friuli and in the northern Dinarides, both, (oblique) thrust and strike-slip faulting is observed (Pondrelli et al., 2006). This overview is by far not complete, but provides a small glimpse into the complex seismic and tectonic activity that is not simply dominated by the main active deformation fronts at the southern and northern margin of the Alps, but also occurs in smaller fault systems across the entire region.


To study the orogenesis of the Alps and related processes like recent seismic activity, mantle dynamics, plate motion and surface processes, the AlpArray initiative was established. In this initiative, more than 35 European institutes joined resources to operate the AlpArray Seismic Network (AASN) (Hetényi et al., 2018), consisting of more than 600 temporary (AlpArray Seismic Network, 2015) and regional permanent stations with an average spacing of <60 km (Fig. 1). For comparison, in

summer 2011, before the AASN planning period started and first additional permanent stations were set up, there were 234 stations in the same area (Hetényi et al., 2018). The AASN is complemented by the dense Swath-D network in the eastern Alps (Heit et al., 2017). The permanent stations of the AlpArray are part of existing European regional networks (RD, GU, CZ, ST, G, CH, OE, MN, HU, GE, RF, FR, IV, BW, SX, NI, TH, OX, see data and code availability). The dense AASN allows studying regional seismicity in new, greater detail and provides the opportunity to perform MT inversions with a constant

station coverage over the entire region. In contrast, many of the previous studies focus on specific regions or seismic sequences within the Alps, therefore not providing a broad overview. Furthermore, many of these studies relied on first motion polarities. First-motion based approaches can be used even for small earthquakes when no surface wave energy is observed. However, the obtained mechanism is only representative for the very first moment of the fracturing process. Therefore, complex rupture processes cannot be represented. This might introduce discrepancies when comparing first motion solutions to MT solutions

(Scott and Kanamori, 1985; Guilhem et al., 2014). Additionally, first motion solutions of small earthquakes are often only based on few data points, which makes it difficult to assess uncertainties.





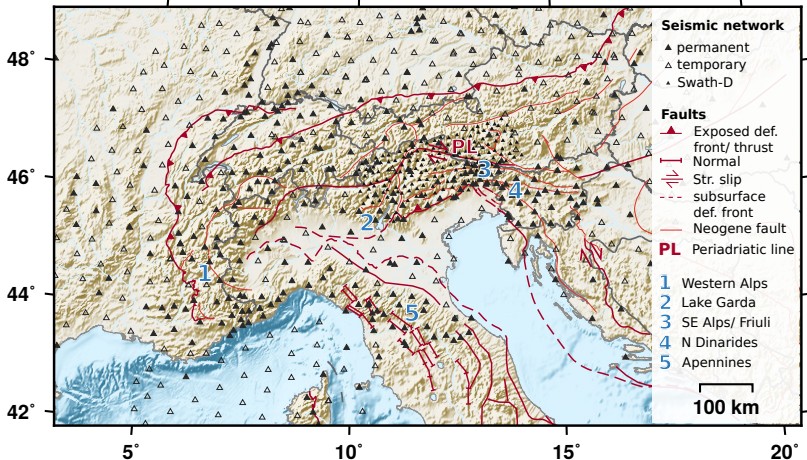

**Figure 1.** Study area and AlpArray seismic network. Subregions that are discussed in greater detail in this study are numbered. Exposed and subsurface faults simplified from Schmid et al. (2004, 2008); Handy et al. (2010, 2015) and Patacca et al. (2008). Topographic data from SRTM-3 (Farr et al., 2007) and ETOPO1 (Amante and Eakins, 2009) datasets).

Despite the limited resolution, first motion polarity approaches are often used in the Alps, where MT inversion is particularly challenging. First, earthquake magnitudes are generally small to moderate, requiring waveform modeling at relatively high frequencies and local distances. Furthermore, structural heterogeneities, site effects and topographic effects hinder full waveform
MT inversions based on 1-D velocity profiles when considering frequencies above 0.1 Hz. Signal-to-noise ratios (SNR) vary significantly across the region due to densely populated areas and environmental conditions (weather and wind, rivers, rock falls, avalanches, etc.). Each study or observatory reporting focal mechanisms uses different inversion tools, input data, as well as distance and frequency ranges. Further, uncertainties are not routinely discussed, which makes it difficult to evaluate published solutions. Uncertainties can be assessed e.g. by performing a grid search over parameters like strike, dip, rake and depth
(Stich et al., 2003; Cesca et al., 2010) or using independent bootstrap chains in the inversions with varying weighting of the input data from different stations (e.g. Heimann et al. (2018); Kühn et al. (2020) and this study). The latter approach provides uncertainties of all inversion parameters (e.g. MT components, centroid location) and helps to identify trade-offs between these parameters.

Studies of focal mechanisms in the Alps have mainly considered DC solutions. Here, based on the dense seismic network,
we also attempt to consider non-DC components. The decomposition of the moment tensor allows studying the seismic source in more detail, including not only pure tectonic dislocations represented by the DC, but also volumetric changes and tensile faulting (e.g. Vavryčuk (2015)).

In this study, we predominately target small to moderate earthquakes (Mw>3.0) that occurred in the Alps or surrounding areas between 01/2016 and 12/2019, based on the operation time of most of the temporary broadband stations of the AASN.
While the temporarily densified network provides a great amount of input data to the single MT inversions, the short time span limits the number of observed earthquakes. We test various set-ups and input types to establish work flows for a homogeneous





and consistent list of MT solutions for the Alps (see Supplement). We attempt to lower the magnitude threshold for inversions compared to routinely reported solutions by optimizing the used methods and by combining different input data types (e.g. time domain full waveforms and frequency domain amplitude spectra). Using the AASN and the bootstrap inversion framework

*Grond* (Heimann et al., 2018; Dahm et al., 2018; Kühn et al., 2020) allows determining most suitable set-ups for source inversions of small to moderate earthquakes within the study area.

After an introduction into the inversion method, we describe the methodological tests that were performed to assess methodological inherent uncertainties. At the same time, we propose guidelines for MT inversion of small to moderate earthquakes in complex tectonic settings. Subsequently, we present the MT solutions that were obtained for the Alpine region and dis-

cuss these with respect to mechanisms, spatial patterns and centroid depths. We discuss different tectonic areas in the Alps systematically, including observations of seismicity, faulting mechanisms and GNSS deformation data.

## 2 Methodology

### 2.1 Moment tensor inversion using Grond

We use the open source software *Grond* for MT inversions (Heimann et al., 2018; Kühn et al., 2020). In a Bayesian bootstrap-

based probabilistic joint inversion scheme, solution uncertainties are retrieved along with the best fitting MT solution (see also Dahm et al. (2018)). We invert for the six independent moment tensor components, the seismic moment, the centroid location and the origin time simultaneously. The objective function is set up in a flexible way to combine different input data types as a weighted sum. In our study, we use combinations of time domain full waveforms, time domain cross-correlations and frequency domain amplitude spectra as input for the inversion. Following the studies of Zahradník and Sokos (2018) and Dahal and Ebel

(2020), we implemented envelopes of time domain waveforms. The misfits of the different input data are combined using an L1 or L2 (or higher) norm. Stations are weighted to account for different epicentral distances (Heimann, 2011). Without applying such a weighting, summed misfits are always dominated by the closest stations which have the highest amplitudes.

Precalculated Green's function data bases are used for rapidly computing synthetic data (Heimann et al., 2019). In our case, we used regional velocity profiles from the CRUST2.0 Earth model database (see http:/igppweb.ucsd.edu/~gabi/crust2.

html, last access June 2020, and Bassin et al. (2000)). The Green's function databases were calculated with the orthonormal propagator algorithm QSEIS (Wang, 1991). Grond selects the appropriate time window, corrects the recorded waveforms for the instrument response and rotates to ZRT coordinate system. Filters are applied and, if specified, waveform attributes as spectra or envelopes are calculated. The inversion is performed in parallel bootstrap chains (here: 100 for normal inversions, 500 for method testing), where individual bootstrap weights are applied to the single station-component-input data type combinations.

Bootstrapping is applied for two reasons: Firstly, to avoid distortions due to few high misfit values resulting from a low SNR at single stations, wrong transfer functions or malfunctioning stations. Further, the bootstrap chains are used to access model parameter uncertainties and trade-offs between the inversion parameters. Each bootstrap chain performs an entirely independent optimization. Along with the best solution with the lowest misfit, *grond* provides a defined number (here: 10) of best solutions of each BS chain, which we call the ensemble of solutions.





**Figure 2.** Example MT inversion result, 2019-06-14, Mw 3.9, NE Italy. (a) Fuzzy beachball illustrating the MT solution uncertainty. (b) Station distribution around the epicenter. (c) Decomposition of the best MT solution into DC and CLVD component. (d) Resolution of the centroid depth, easting and northing relative to the starting position. (e) Probability density functions (PDF) showing the resolution of the six independent MT components $m_{xy}$ normalized by the seismic moment $M_0$. (f) Examples of waveform and spectral fits at three stations and three components. Red and black lines indicate synthetic and recorded waveform data, respectively. Station name, azimuth and distance to the epicenter above each column. Numbers within the panels describe the time window and the frequency band.





Fig. 2 presents a selection of plots provided by the inversion software to assess the solution robustness, in this case for
the 2019-06-14 thrust faulting earthquake in northern Italy (Mw 3.9). Fig. 2a shows the fuzzy MT, which is an illustration of
the MT uncertainty. It is composed of the superimposed P radiation pattern of the ensemble of solutions from the bootstrap
chains. If the variability of the ensemble solutions is small and hence the uncertainties are small (as seen here), the fuzzy
plot has clearly separated black and white quadrants. The red lines indicate the solution with the lowest misfit. Other plots

show the station distribution (Fig. 2b) and the decomposition of the best deviatoric MT solution into the DC and the CLVD
component (Fig. 2c). Fig. 2d shows the distribution of the centroid locations obtained from all ensemble solutions and Fig. 2e
depicts the resolution of the six independent MT components in form of probability density functions. As typical for shallow
events inverted using surface waves, the $m_{nd}$ and $m_{ed}$ components are not as well resolved as the other components (Cesca
and Heimann, 2018; Bukchin et al., 2010; Valentine and Trampert, 2012). Finally, Fig. 2f shows examples of waveform and

frequency spectra fits of Z, R and T component traces at three stations. Further plots, e.g. showing parameter trade-offs and the
evolution of misfits and optimised parameters during the inversion, are not provided here.

The selection and joint inversion of waveform attributes can improve the stability and goodness of solutions. In the following,
we want to point out advantages and drawbacks of the waveform-based input data types, which are used in the subsequent

methodological tests: time domain (TD), frequency domain (FD), cross-correlation and envelopes.

**TD full waveform fitting:** In time domain, the misfit between a selected time window of a seismic trace and a synthetic trace
in a defined frequency range is computed as the normalized sum of sample misfits. Time shifts are allowed and regulated with
a penalty function. For regional MT inversion surface waves are commonly considered in full waveform approaches (e.g. Rit-

sema and Lay (1995), Minson and Dreger (2008), Sokos and Zahradnik (2008) and Dahm et al. (2018)). The frequency band is
magnitude dependent. While at low frequencies effects of the velocity model and topography are minor, at higher frequencies
the SNR is usually better. At regional distances, often magnitude-dependent frequency bands below 0.1 Hz are used to consider
Rayleigh waves and Love waves, which have particularly simple waveforms at regional distances (Ritsema and Lay, 1995).
However, in case of very shallow sources, the resolution of the $m_{xz}$ and $m_{yz}$ components of the MT are limited when using

surface waves (Bukchin et al. (2010); Valentine and Trampert (2012); Cesca and Heimann (2018), see also Fig. 2). Relying
on time domain fitting only, time shifts, noisy data or distorted amplitudes can hinder finding stable initial inversion solutions.
It has proven to be helpful to combine time domain full waveform fitting with other input data types like frequency domain
amplitude spectra which are often less affected by these issues.

**FD amplitude spectra fitting:** Real valued amplitude spectra of recorded Love and Rayleigh waves carry all information
necessary to resolve the geometry of the MT, while neglecting the phase information and dispersion (e.g. Mendiguren (1977)).
This means two MT solutions with common nodal planes but opposite polarities model the amplitude spectra equally well and
additional information from first-motion polarity readings or time domain waveform fitting is needed to resolve this ambiguity
(e.g. Cesca et al. (2010), Heimann (2011)). The misfit is computed as the misfit between amplitude spectra of recorded and





synthetic waveforms in a selected frequency range and time window. Compared to fitting full waveforms in time domain, more conservative, less exact time windows can be selected. Cesca et al. (2010, 2013) propose a multistep approach to stepwise combine the fitting of amplitude spectra and displacement waveforms to subsequently obtain point source parameters, the centroid location and kinematic source parameters. Compared to full waveform fitting, amplitude spectra inversion methods are less sensitive to trace misalignments and phase shifts resulting from coarse or erroneous velocity models (Cesca et al.,

2010, 2013; Domingues et al., 2013). In the subsequent tests, we do not use a step-wise inversion, but use amplitude spectra and time domain full waveforms or cross-correlated waveforms simultaneously.

**Cross-correlation waveform fitting**: In the cross-correlation based fitting of full waveforms, the amplitudes of recorded and synthetic traces are normalized. The inversion searches for the maximum cross-correlation value for the selected time window

in time domain, basically fitting the phase shift (Stähler and Sigloch, 2014; Kühn et al., 2020). Cross-correlations help to constrain the centroid location and centroid time in a joint inversion. We allow for small time shifts, regulated with a penalty function, to compensate for imprecise velocity models. Time shifts need to be small compared to the frequency range in order to avoid mismatching phases. Due to the amplitude normalization, this method is sensitive for patterns in the waveforms, while it is not influenced by gain errors or site effects. Magnitudes cannot be resolved. Cabieces et al. (2020) used cross-correlation

fitting in their MT inversion for ocean bottom stations, where absolute amplitudes could not be modelled due to the unknown coupling to the ground. When using cross-correlations to fit time domain waveforms, both, frequency bands as well as time windows need to be selected carefully to avoid mismatching phases.

**Envelope fitting**: In our study, we compute the waveform envelopes by convolving the squared time series using the fast

Fourier transform with a Hanning taper. This smoothens and therefore simplifies the waveforms. Hensch et al. (2019) used non-smoothed envelopes in the same inversion routine in combination with amplitude spectra, spectral ratios and time domain waveforms. Envelopes, especially smoothed ones, are less influenced by small unmodelled time shifts or noisy data compared to full waveforms. Fitting envelopes of seismic waveforms can be helpful in case of using high frequency bands, simplified velocity models and increased noise levels. Zahradník and Sokos (2018) stress that due to the simplification of the waveforms,

the results of envelope-based inversions have a limited precision and results need an even more careful inspection of uncertainties and resolution. However, if body waves are considered, envelopes can especially help to constrain P and S phase arrivals, and thus the centroid time and location. Since the envelopes are based on absolute amplitudes, they need to be combined with a method providing polarity information. Zahradník and Sokos (2018) and Dahal and Ebel (2020) have shown that envelopes can be used to derive focal mechanisms for M<4 events in case of unfavorable settings like sparse networks, for which full

waveform fitting is not feasible. In both studies, the envelopes are combined with P-polarities of one or more nearby stations to resolve the polarity ambiguity.





## 2.2 Methodological tests

We perform methodological tests using recorded seismograms and synthetic data to investigate the resolution capacities, re-
quirements and limits of MT inversion in the Alpine region. We use subsets of representative earthquakes that occurred between
2016 and 2019 in the Alps. The tests are particularly computationally demanding, as every single inversion of each test is run
in 500 bootstrap chains. The number of events in each test depends on the number of tested parameters. The proposed tests can
be used as a guideline for assessing the feasibility of MT inversions in other study areas with moderate seismicity.

### 2.2.1 Double-couple, deviatoric and full moment tensor inversions

We study the stability and resolvability of non-DC components by performing full, deviatoric and pure-DC MT inversions
for a subset of 32 representative earthquakes of the AlpArray data set (Mw 3.2-4.2). We use a passband of 0.02-0.07 Hz
and fit time domain full waveforms and frequency domain amplitude spectra simultaneously. We compared the pure-DC, the
deviatoric and the full MT obtained for each earthquake with respect to the fit of the recorded data and to the uncertainties of
MT components and centroid locations. Subsequently, we statistically evaluate those solutions, for which a low misfit between
synthetic and observed waveforms was achieved with all three inversion types. Four events with Kagan angles > 60° between
full and deviatoric solutions were removed, since they were not well resolved.

Fig. 3a and b show the ratio of DC and non-DC components of the full and deviatoric MT inversions. The deviatoric
inversions result in 0-40% CLVD components for 70% of the events and in CLVD components of >50% for 19% of the events.
In case of full MT inversions, we find significant isotropic components of >30% in case of one third of the earthquakes. Fig.
3c indicates that the non-CLVD components of the test events scatter significantly. It is clearly visible that many events with
shallow depths (dark colors) are located in the upper right and lower left quadrants of the Hudson plot, indicating isotropic
and CLVD components of opposite signs. Cesca and Heimann (2018) showed that for shallow depths, isotropic and CLVD
components often appear indistinguishable. Further below, we discuss this observation comparing forward calculated synthetic
waveforms for one example event.





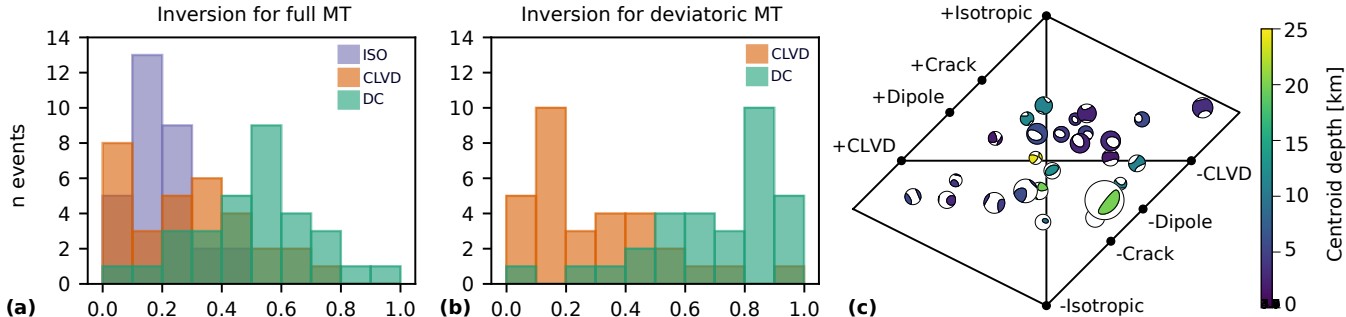

**Figure 3.** Histograms showing the decomposition of the full (a) and deviatoric (b) MT inversion results into isotropic (ISO), compensated linear vertical dipole (CLVD) and double couple (DC). Each bar of 10% width (x-axis) indicates for how many test earthquakes (y-axis) the proportion of decomposition is found. For example, an isotropic component of 10-20% is found for 13 test events in the full MT inversion. (c) Hudson plot showing non-DC components of individual events. Four events with Kagan angles > 60° between full and deviatoric solutions were removed, since they were not well resolved. Color represents depth, beachball size magnitude.

The DC component is representing the purely tectonic shear dislocation (e.g. Miller et al. (1998); Julian et al. (1998); Cesca et al. (2013)), therefore, it is crucial to resolve this component unambiguously. We compare the DC component that we obtain from the decomposition of the deviatoric and the full MT with the pure-DC inversion result by computing the smallest rotation angle (Kagan angle, Kagan (1991)) between them to assess the stability of the DC components (Fig. 4). Rotations below 30°

are generally accepted as representing very similar mechanisms, while a Kagan angle ≪60° is still described as corresponding (Pondrelli et al., 2006; d'Amico et al., 2011). 70% of the earthquakes have a very stable DC, with Kagan angles below 30° between the three solutions. In case of about 10% of the events, a Kagan angle ≥60° is found. These larger deviations result predominantly from large non-DC components in the full inversion result (in >70% of these events). In these cases, the CLVD combined with the isotropic component shows orientations similar to the DC component of the pure-DC and the deviatoric

inversion result. Therefore, the resulting focal sphere is similar, while the DC component deviates from the pure-DC inversion result. Overall, the results of this test indicate that the DC component is in most cases very well resolved, independently of allowing for a CLVD and an isotropic component.




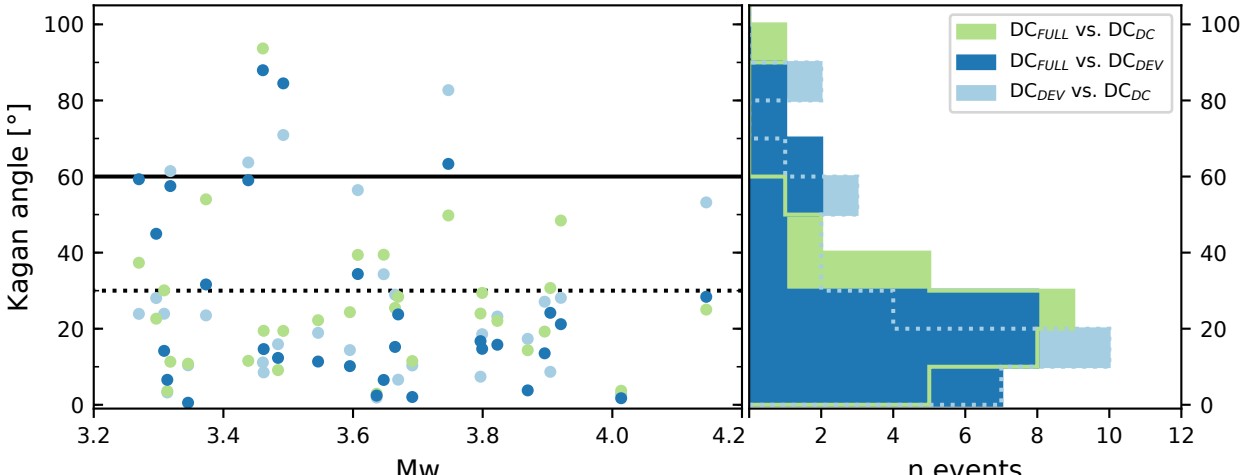

**Figure 4.** (left) Kagan angle between DC of pure-double-couple, deviatoric and full MT solutions for 30 earthquakes, Mw 3.3-4.5. The dashed and solid lines indicate Kagan angles of 30 and 60°, respectively, indicating levels of high agreement and corresponding mechanisms. (right) Histograms showing the distribution of Kagan angles between the DC of pure-DC, deviatoric and full MT inversion solutions.

The isotropic source component resolves volumetric changes, including processes as explosions, cavity collapses, fluid movement or ruptures on nonplanar faults (e.g. Sileny and Hofstetter (2002); Ford et al. (2009); Miller et al. (1998); Minson and Dreger (2008). The CLVD component is often described as the residual radiation to the best DC without geological interpretation (Dahm and Krüger, 2014), it is however required for a mathematically complete decomposition (Vavryčuk, 2015). Large CLVD components are often explained by noisy data, a simplified or incorrect velocity model, neglected 3-D wave effects or insufficient station coverage (e.g. Panza and Saraò (2000); Cesca et al. (2006)), but can also be interpreted physically in combination with an isotropic component of the same sign as a product of tensile faulting (Vavryčuk, 2015). Non-DC components are also used as an indicator of anthropogenic seismicity (Dahm et al., 2013; Cesca et al., 2013; Lizurek, 2017).

Despite frequent geological interpretations which propose fluid movements or tensile processes, various studies show that resolving non-DC components in MT inversion is particularly difficult. Seismic noise and inaccurate Green's functions may result in large non-DC components. Trade-offs between hypocenter location or depth and isotropic component have been observed (e.g. Dufumier and Rivera (1997), Panza and Saraò (2000), Křížová et al. (2013), Kühn et al. (2020)). Non-DC components must therefore be evaluated carefully with respect to tectonic processes (Lizurek, 2017). Synthetic tests can help to identify which non-DC components can be considered statistically significant (Panza and Saraò, 2000).

In the following, we present a more detailed analysis of an exemplary earthquake with a significant non-DC components: 2019-05-28, Mw 3.9, close to Lake Geneva, France (Fig. 5a). Fig. 5b depicts the MT decompositions of the MTs obtained with a pure-DC, a deviatoric and a full MT inversion. All three inversions were performed using the same inversion set-up





(full waveforms and amplitude spectra, Z,R and T components, 73 stations, 0.02-0.07 Hz). The DC component is similar for all inversion types, but the deviatoric and full inversion results indicate significant non-DC components.

To investigate whether the resolved non-DC components are unambiguous, we forward model synthetic waveforms of the three MT solutions recorded at fictional receivers in 250 km distance in azimuthal steps of 1° (Fig. 5a). We use a bandpass-filter of 0.02-0.1 Hz, which is even wider than the frequency range used in the MT inversion (0.02-0.07 Hz) to assess the similarity of the entire modelled surface wave trains. By cross-correlating the forward modelled waveforms, we find that the full and deviatoric sources produce very similar waveforms on all seismometer components in all backazimuthal directions (Fig. 5d). Further, the maximum amplitudes between deviatoric and full solution differ only slightly. This indicates that the non-DC

component can be comparably well represented by a CLVD or by a combination of an isotropic plus a CLVD component.

A comparison of the forward modelled waveforms from a pure-DC solution with a full or deviatoric solution shows very high correlations in all azimuthal directions on the T components (Fig. 5d, lower panel). Neither the CLVD nor the isotropic component is influencing the transversal Love wave. On R and Z components, the resulting waveforms show cross-correlations below 0.9 in strike-direction only (Fig. 5d, upper and middle panel). This indicates that in case of this event, without stations

covering this ray path direction, we cannot resolve the difference between a pure-DC MT and a full or deviatoric one.

The true azimuthal coverage of seismic stations is much denser to the NE and E than in strike direction (Fig. 5a). Forward modelling the waveforms of the 73 used stations, results in a similar, but less well resolved pattern compared to Fig. 5d. The uneven azimuthal distribution and the lack of stations in strike directions hinders the unambiguous identification of non-DC components.

This example shows that whenever we investigate large non-DC components in a deviatoric or full MT inversion, we have to asses carefully teh resolution and the validity of the results. The synthetic tests indicate that including full waveforms of body waves at higher frequencies in the inversion clearly helps to improve the resolution of non-DC components. However, due to the station spacing, the relatively high noise level and the low resolution of crustal velocity models, we cannot use higher frequencies for most events in this study. Following our findings, we report deviatoric MT inversions in the result section and

only perform inversions for the full MT in case of large non-DC components for comparison.

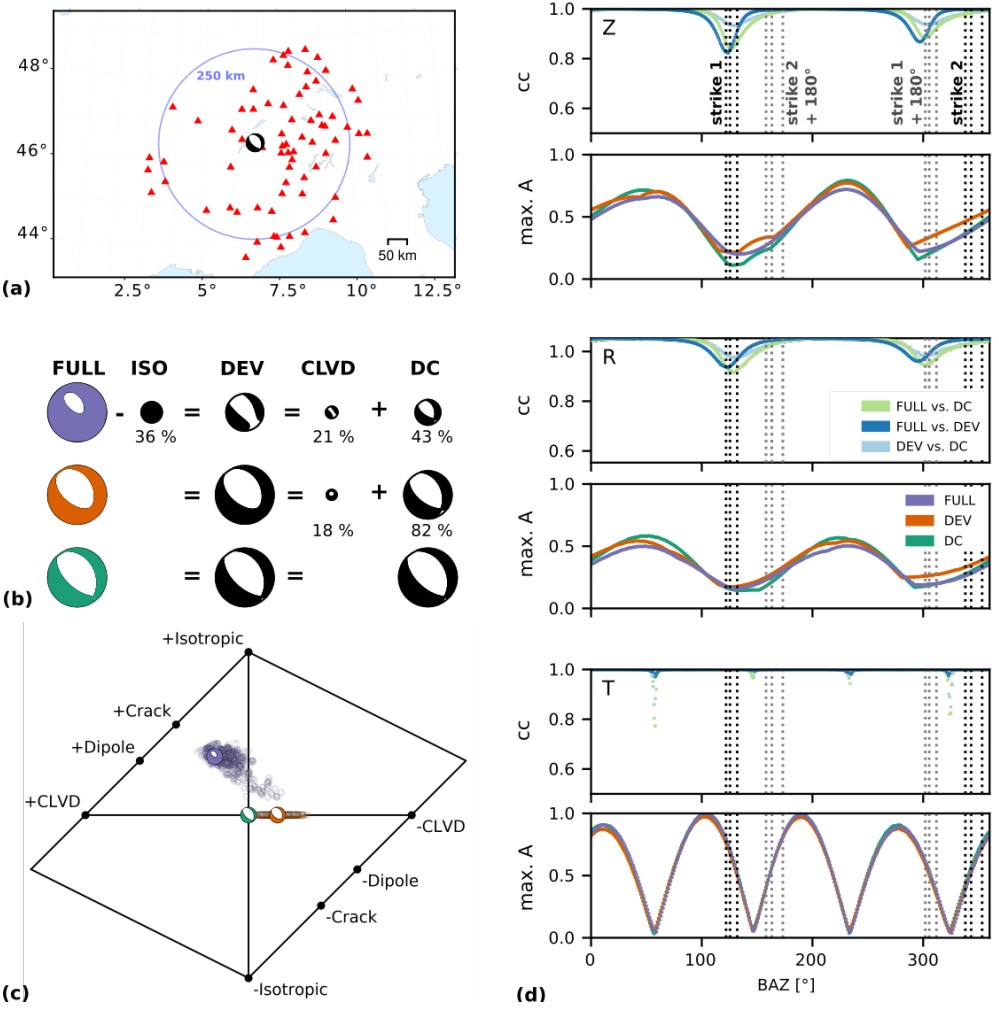

**Figure 5.** Earthquake close to Lake Geneva (France-Swiss border region), Mw 3.9, 2019-05-28 08:48:06. (a) Seismic network used for the MT inversions (73 broadband sensors, red triangles) and synthetic network (blue dots, in 1° steps). (b) Decomposition of the full (purple), deviatoric (orange) and pure-DC (green) MT inversion solution with the smallest misfit. (c) Hudson plots showing the ensemble of solutions for the full and the deviatoric MT inversions, colors as in (b). Larger symbols depict best solutions of b. (d) For the three MT solution (full, deviatoric and DC) synthetic data were forward computed for the fictional network shown in (a). For each component (Z,R,T), the first row shows the maximum cross-correlation values of the three synthetic traces at each station (BP filter 0.02-0.1 Hz). The second row shows the maximum absolute amplitude at each backazimuth, normalized over the three solutions. The dashed lines indicate the strike 1 and 2 directions of the two nodal planes of full, deviatoic and DC solution and their 180° equivalent.



### 2.2.2 Magnitude-distance relation

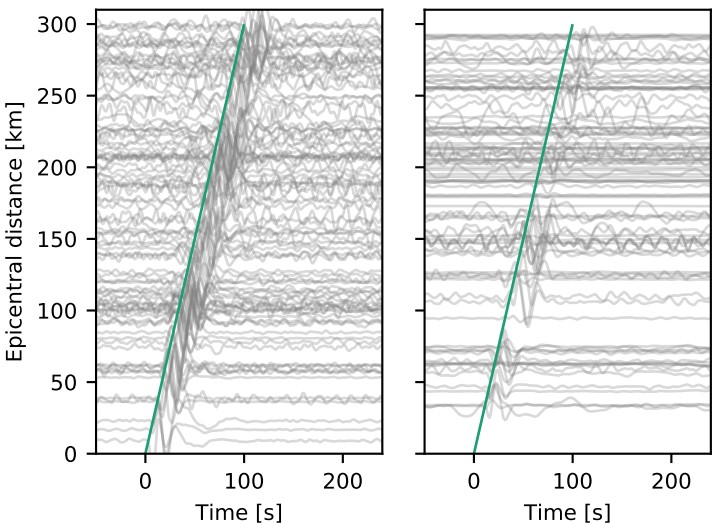

**Figure 6.** Vertical component seismograms of permanent AASN stations, sorted by distance. (left) 2017-03-06, Mw 4.1, Switzerland, (right) 2017-10-27, Mw 3.6, France. Time in seconds after origin time. Waveforms bandpass filtered between 0.02 and 0.07 Hz. Rayleigh waves dominate the seismograms in this frequency range with an average phase velocity of 3 km/s (green line).

The distance range in which stations can be used for MT inversion strongly depends on the event magnitude. While we use an epicentral radius of less than 100 km for the smallest events with magnitudes Mw 3.1-3.3, epicentral distances may be as large as 300-400 km for the largest events with magnitudes greater Mw 4.0. For events between Mw 3.5-3.7 and Mw 3.8-3.9, we use distances of up to 160 km and 200 km, respectively. This results in different inversion set-ups: the number of available stations varies between less than 10 stations for the Mw 3 events to above 80 stations for the largest events. Fig. 6 illustrates this relation. The left panel shows waveforms of an Mw 4.1 event in Switzerland at distances of up to 350 km. Even though the SNR decreases with distance, a distinct Rayleigh wave can be seen. The second event from 2017-10-27, France, has a magnitude of Mw 3.6. For distances larger than 160 km, SNR are very low for most stations. We did not remove stations with generally high noise levels from the plot to illustrate that a careful rejection of very noisy and disfunctional stations is required. We apply the toolbox *AutoStatsQ* in advance to identify seismic stations that are misoriented, have errors in their metadata or gain problems (Petersen et al., 2019).

### 2.2.3 Frequency ranges and input data type

In previous studies, dependencies of the MT inversion results on the inverted frequency band have been observed and multistep inversion workflows including several frequency bands were proposed (e.g. Barth et al. (2007)). In order to find the best combination of frequency ranges and time domain or frequency domain input types for the MT inversion, we selected a subgroup





of 13 earthquakes. These test events span a magnitude range of Mw 3.3 to 4.1 and are therefore considered representative. We perform MT inversions using eight different combinations of input data types (Fig. 7): time domain full waveforms (td), frequency domain amplitude spectra (fd), cross-correlations of time domain full waveforms (cc), waveform envelopes and

combinations thereof. The input data is filtered using nine different bandpass filters with passbands between 0.01 and 0.7 Hz (Fig. 7). We compare the uncertainties of the resolved MTs in order to find the most appropriate parameter settings for our study and future MT studies in the Alps or similar settings.

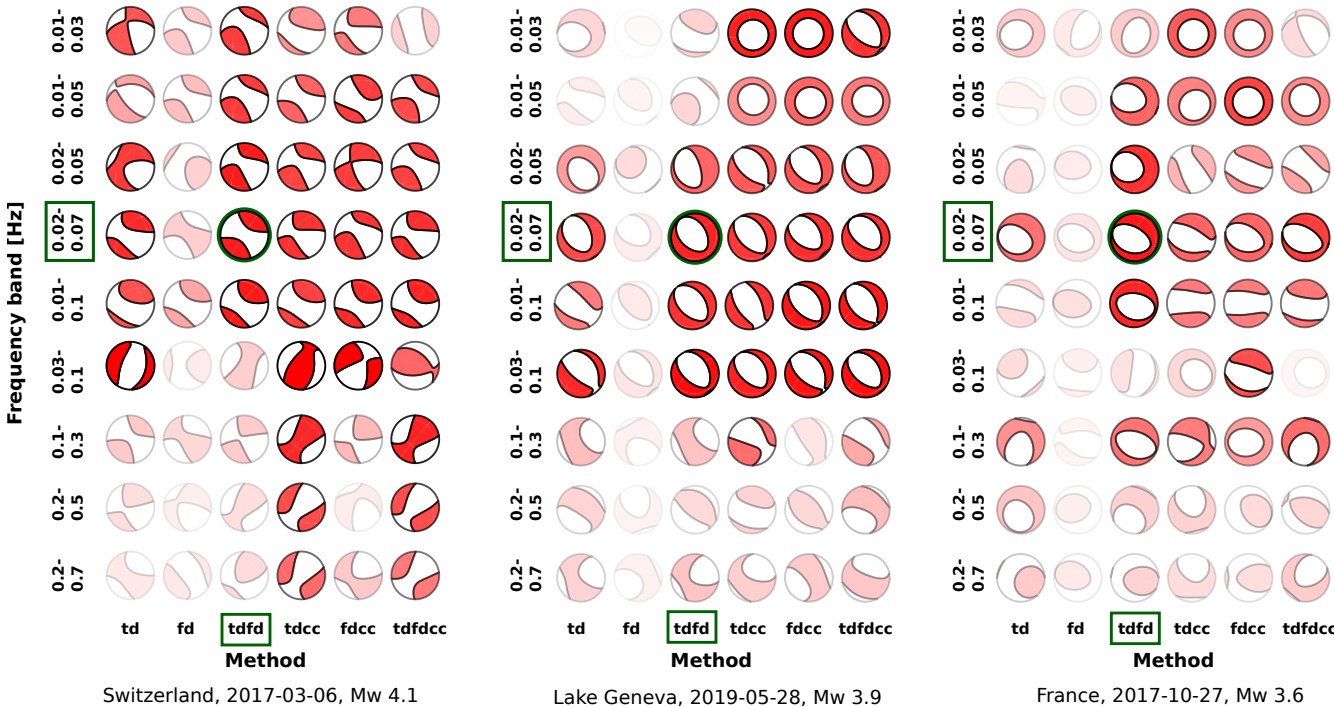

**Figure 7.** Testing of input data types and frequency ranges for three earthquakes with magnitudes between Mw 3.6 to 4.1. Abbreviations: td - time domain full waveforms; fd - frequency domain amplitude spectra; cc - cross-correlation fitting of full waveforms; tdfd, tdcc, fdcc and tdfdcc are combinations of these. Color intensity represents summed uncertainty of MT components. A combination of fd and td in a frequency band between 0.02-0.07 Hz yields the best results for Mw > 3.3 (marked in green).

In Figures 7, we show the results for three exemplary events. The color intensity of each focal sphere represents the summed

standard deviations of the six MT components derived from the ensemble of solutions of the bootstrap chains. Intense colors represent stable solutions with low uncertainties. The first event is a Mw 4.1 earthquake in Switzerland (Fig. 7). Due to the high magnitude, the MT inversion results are stable over frequency bands ranging from 0.01-0.03 Hz up to 0.01-0.10 Hz for all input data types. The MT is not well resolved when filtering using a passband of 0.03-0.1 Hz or higher. A similar behaviour is observed for the second example, a Mw 3.9 normal faulting event close to Lake Geneva. The MT is very well resolved using




bandpass filters covering the intermediate passbands between 0.02 and 0.1 Hz. In contrast to the first event and corresponding to the lower magnitude, the resolution is worse when frequencies between 0.01-0.02 Hz are included, while the frequency band between 0.03-0.1 Hz still leads to satisfying results. In general, the higher the frequency band, the lower the stability of the ensemble of solutions due to the simplified 1-D velocity model, site effects and increased noise levels.

    In case of the third example (Fig. 7), a Mw 3.6 earthquake from France, the MT solutions vary substantially. This illus-
trates the need for a careful selection of appropriate methods and frequency ranges and the analysis of the uncertainties of MT inversions. For both, the higher frequency ranges (from 0.03-0.1 Hz and higher), as well as in the lowest frequency bands (0.01-0.03 Hz and 0.02-0.05 Hz), surface waves have insufficient SNRs. Stable results for most input types are obtained in the frequency band 0.02-0.07 Hz, in which surface waves are more distinct. A visual inspection of the recorded waveforms of various events with magnitudes of Mw 3.4-3.9 confirms that surface waves have highest SNRs for periods between 0.02
and 0.05 Hz. Extending the passband to 0.02-0.07 Hz helps to avoid mismatching monochrome phases in the inversion process.

    Comparing the different input data types for all thirteen test events, we find that a combination of frequency domain amplitude spectra and time domain full waveform fitting (tdfd in Fig. 7 provides more stable results than relying on time domain waveform fitting alone. The high uncertainties of the frequency domain amplitude spectra fitting alone (fd) result from the
unresolved polarity. The geometry of the nodal planes can still be determined. For most events, the other combinations (tdcc, tdfccc, fdcc) provide more stable results compared to using only time domain full waveforms (td). However, compared to the tdfd combination, they do not further improve the stability of the solution.

    In addition to the presented tests of input data types, we tested waveform envelopes (Supplement Fig. S1). In order to resolve the polarity of the mechanisms, the envelopes are combined with time domain full waveforms or cross-correlation
fitting of waveforms at close-by stations. This is a reasonable setting for weak events, where full waveforms may be of such low amplitudes that they can only be fitted at closer stations while envelopes of more distant stations may still be of use. We find that in case of intermediate or large magnitude test events (Mw≥ 3.6), the resulting MT is well recovered, although uncertainties are larger than with a time domain-frequency domain combination. In case of smaller events, where time domain-frequency domain combinations might fail, the envelopes may stabilize the inversion. The applicability of the combination of
envelopes and close-by time domain traces depends on the data quality of the closest stations and on the careful selection of the frequency range and the smoothing of the waveform envelopes.

    Following the results of our methodological tests, we routinely use a combination of frequency domain amplitude spectra and time domain full waveform fitting in a frequency band of 0.02-0.07 Hz for earthquakes with Mw >3.5. In case of smaller magnitude earthquakes, we additionally perform inversions using a frequency range of 0.03-0.10 Hz. We observe that in case
of low magnitude earthquakes the initial local magnitudes can differ significantly from our moment magnitude estimates. Further, the availability of stations with a good SNR does not only depend on the event magnitude, but also on noise conditions and damping along the travel path. It is therefore necessary to adapt the approach to the individual earthquakes, but the two frequency ranges constitute reasonable guidelines.





### 2.2.4 Station coverage

The dense AASN provides an excellent azimuthal distribution of seismic stations for moderate to large earthquakes in the Alps. We take advantage of the large number of stations in the AASN to investigate how the stability of the MT inversion is influenced by gaps in the azimuthal station distribution around an earthquake. This allows simulating uncertainties of MT solutions in the marginal areas of the AASN, but the results also apply to other locations and networks (e.g. close to subduction zones). Most generally, within the AASN larger event-station distances can be taken into account for larger magnitude earthquakes and

therefore both the number of stations and the azimuthal station coverage increases. In contrast, individual malfunctioning stations may already result in large azimuthal gaps for low magnitude earthquakes located within the AASN.

Fig. 8 shows the fuzzy MTs (right panels) for decreasing azimuthal coverage of seismic stations (left panels) for three exemplary events. In case of the largest event (Mw 4.1), the solution is very stable when seismic stations cover at least an azimuthal range of 90°. In case of an even smaller coverage, the mechanism rotates slightly depending on the azimuthal

direction of the remaining stations. In case of the Mw 3.9 event, the uncertainties of the solutions increase with decreasing station coverage. Two examples in which the inversions were done with stations covering only an azimuthal range of 45° show significant differences between the resulting focal mechanisms. When only considering the fuzziness of the two focal mechanism plots, both ensembles of solutions seem to be well resolved and stable. This indicates that the amount and variability of input data is not sufficient to resolve the MT unambiguously.

For the smallest earthquake (Mw 3.5), the resulting MT solutions vary even more. In case of the inversions with a station coverage of 90°, the variability among the ensembles of solutions is high and depends on the location of the 90° quadrant covered with stations. It is worth noticing that even with a small number of stations covering a small azimuthal range, it is possible to resolve a MT under favourable geometrical conditions. When stations are located in strike direction and cover both, tensional and compressional quadrants, they may resolve the MT correctly even when covering only 45° (Fig. 8, Mw 3.5 event,

5th row and last row).

We conclude that in case of larger earthquakes with a high SNR and a sufficient number of stations at different epicentral distances even a limited a azimuthal coverage does not necessarily pose a problem, but lower magnitude earthquakes usually require a better azimuthal station coverage. In regard to a semi-automated MT inversion workflow, we implemented an optional minimum station distribution threshold. Based on our results and since we do not assume an a-priori known strike direction, we

limit the inversions to earthquakes with an azimuthal coverage above 90° but thoroughly evaluate all results with a coverage below 180°.





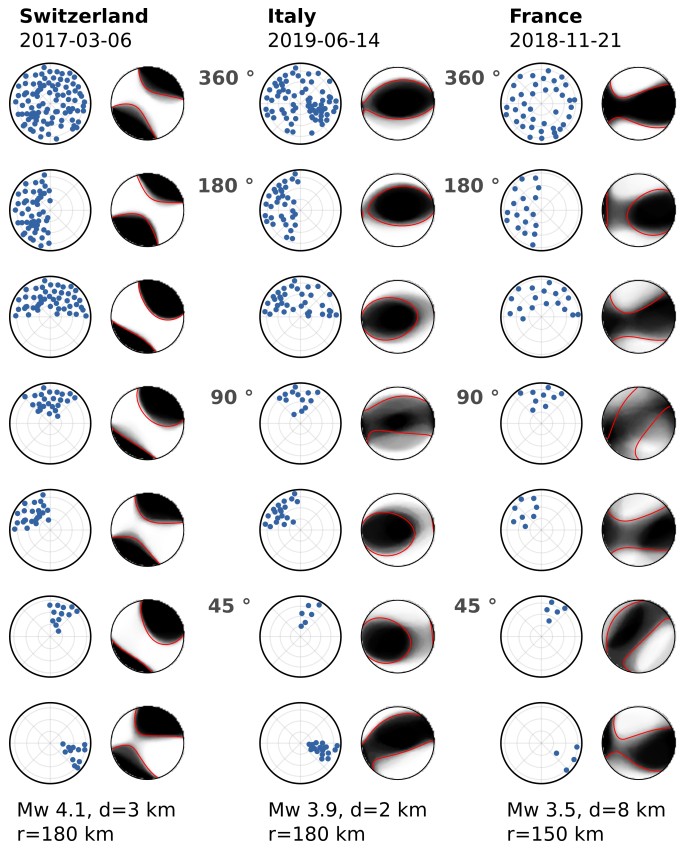

**Figure 8.** Resolution of the deviatoric MT depending on the azimuthal station coverage for three earthquakes with magnitude $Mw$, centroid depth $d$ and event-station radius $r$ given below (left, middle and right panels). The station coverage (blue dots in 1st, 3rd, 5th column) decreases from top to bottom as indicated in between the event-columns. The fuzzy MTs show the solution stability. They are composed of the superimposed P radiation pattern of the ensemble of solutions from the bootstrap chains.

## 3  CMT solutions for the Alpine region, 2016-2019

Based on our methodological tests, we use a combination of time domain full waveforms and frequency domain amplitude spectra as input data for the centroid MT inversion for earthquakes larger than Mw 3.0. We choose a frequency range of 0.02 to
0.07 Hz for a first inversion of each event. Depending on the event magnitude, the maximum epicentral distance varies between 80 and 300 km. In case of poor fits, we slightly increase the frequency bands (0.03-0.1 Hz for Mw<3.3) for smaller events and decrease it for the larger events (0.02-0.05 Hz for Mw>4.2). Deviatoric inversions were generally favored over full moment tensors, since we demonstrated that often the isotropic and CLVD components can not be distinguished reliably. In addition, no volume changes are expected to accompany small earthquakes in the seismotectonic setting of the Alps. We obtained deviatoric
MT solutions for 75 earthquakes occurring between 01/2016 and 12/2019 in the wider Alpine region, for which we determine moment magnitudes between Mw 3.1 to 4.8 (Fig. 9, Table in the supplement). While we were able to compute stable MTs for





most earthquakes from regional catalogs with local magnitudes larger Ml 3.3, we resolved only thirteen MTs for earthquakes with local magnitudes between Ml 3.1 and 3.3, corresponding to one third of the events in this magnitude range. Low SNR in the tested frequency bands covering frequencies between 0.02 and 0.5 Hz and less available stations hindered successful
inversions for the other small earthquakes. Furthermore, we realised that a station spacing of about 60 km is not sufficient for small earthquakes (Mw<3.3) in case a part of the data is rejected due to quality issues.

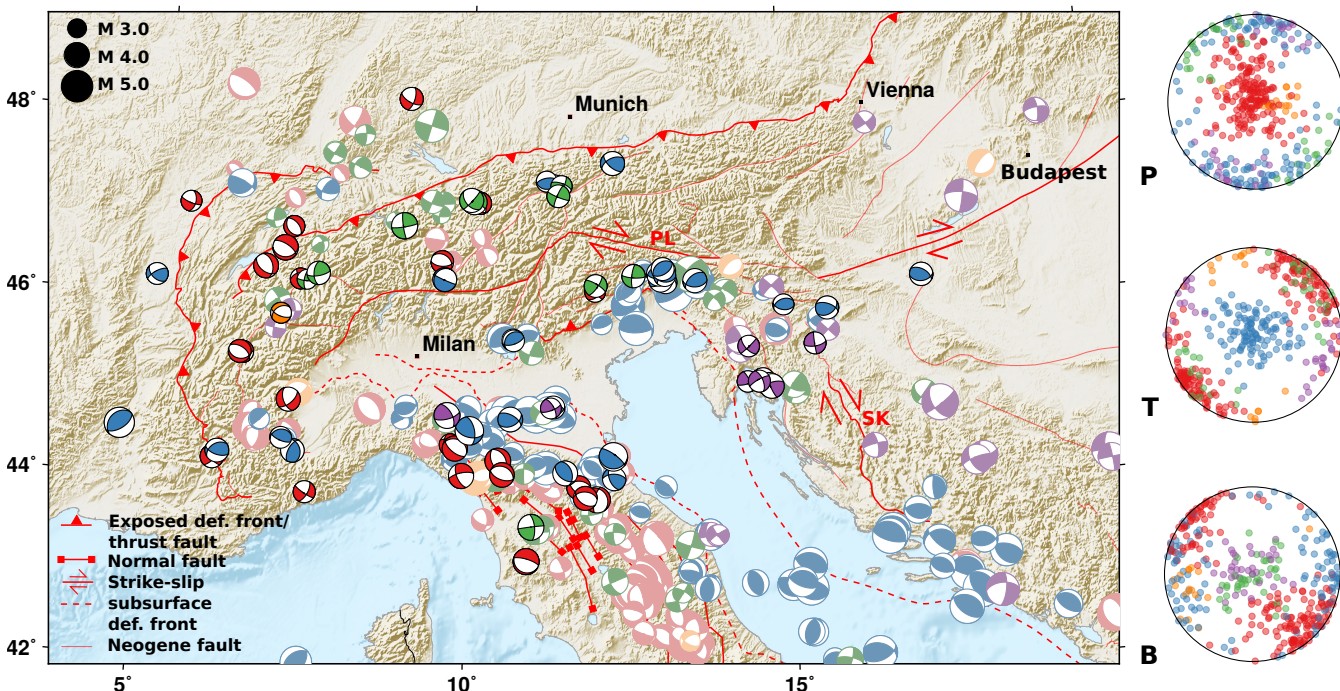

**Figure 9.** Moment tensor inversion results from 01/2016 to 12/2019 (focal spheres with black lines) along with MTs from 1983-2015 from bulletins of gCMT, GEOFON, INGV, SED, EM-RCMT and ARSO (lighter colors). Similar colors represent clusters of comparable mechanisms (see discussion). Exposed and subsurface faults simplified from Schmid et al. (2004, 2008); Handy et al. (2010, 2015) and Patacca et al. (2008). 'PL' marks the Periadriatic line, 'SK' the Split-Karlovac Fault. (right) Pressure (P), tension (T) and null (B) axis of all focal mechanism. Topographic data from SRTM-3 (Farr et al., 2007) and ETOPO1 (Amante and Eakins, 2009) datasets.

For about 40% of our MT solutions, no MTs solutions were available from regional observatories (INGV, GEOFON, EM-RCMT, SISMOAZUR, SED, ARSO). For the other earthquakes, we obtain similar MT solutions, with a median deviation Kagan angle of 21° (mean 24°). For the largest events, the deviation is 15°. Our mechanisms coincide well with long-term
seismological and tectonic observations. Thrust faulting related to the NS compression is mainly observed in the SE Alps, strike-slip faulting is observed in the northern Dinarides and along the northern Alps and normal faulting is observed in the NW Alps (Fig. 9). The resolved centroid depths within the Alps range from about 2 to 15 km, pointing at a shallow seismic activity within the mountain range. 80% of the studied events have depths shallower than 10 km (see also Discussion 4.2.2). A comparison of our inverted centroid depths to the depths published in the event catalogs is limited for two reasons. First,



the event depths were fixed in some of the published moment tensor inversion results, and second, the depth estimates differ significantly among the different catalogs. We find a good correspondence with less than 3 km difference for >60 % of the events to at least one of the published solutions. For another 26% we report differences between 3-5 km.

In the following discussion, we compare our results to approximately 350 moment tensors of earthquakes occurring before 2016 reported by gCMT, GEOFON, INGV, EM-RCMT, SED and ARSO (Fig. 9). Additionally, we evaluate published earth-
quake catalogs and GNSS strain data, to put our results into the seismo-tectonic context and draw a more detailed picture of the seismic activity in the study area.

## 4   Discussion

### 4.1   Dominant mechanisms and the regional stress regime

Within the Alps, we observe four dominant groups of focal mechanisms (Fig. 9): Roughly E-W striking thrust faulting is
observed in the south-eastern Alps (northern Italy) and at the central northern margin of the mountain range (blue focal spheres). A group of similar strike-slip faulting earthquakes are aligned parallel to the northern deformation front of the Alps (green focal spheres, Fig. 9). A second group of strike-slip mechanisms is situated SE of the Alps (purple focal spheres). NW-SE striking normal faulting events are found in the NW Alps (red focal spheres), while mechanisms are more heterogeneous in the SW Alps.
We used a clustering algorithm (Cesca, 2020) based on the Kagan angles between all focal mechanisms obtained in this study and reported by gCMT, GEOFON, INGV, SED, EM-RCMT and ARSO to define classes of similar mechanisms (Fig. 9). The clustering tool uses the DBSCAN clustering algorithm (Ester et al., 1996), which is relying on two parameters, the maximum acceptable similarity distance (eps) between two events, here eps=0.14, and the minimum number of neighbouring items (nmin), here nmin=6. We choose a rather large eps value to emphasize patterns of general similarity between mechanisms.
In a second step, we assign all remaining earthquakes to the cluster to which they have the smallest rotation angle.

The thrust faulting events and most strike-slip faulting events in the Alps show typical NNE-SSW to NW-SE oriented pressure (P) axes, in accordance with the regional compressional stress regime (Fig. 9 and Fig. 10). Strike-slip events in the northern Dinarides and the SE Alps, have N-S to NE-SW oriented P-axes and E-W to ESE-WNW oriented tension (T)-axes. In contrast, the normal faulting events in the NW Alps have NE-SW oriented T-axes (Fig. 9, 10). This points at a local change
in the stress regime. While focusing on the Alps, we also report some MT solutions for the Apennines. Normal faulting, with a similar orientation as in the NW Alps, is dominant along the central arc of the Apennines. The normal faulting events of the NW Alps are located within the strike direction of those in the Apennines. However, despite the similarity of mechanisms, the depths of the events in the NW Alps are significantly shallower (Fig. 11). Further, NW-SE striking thrust earthquakes are dominant along the north-eastern margin of the mountain range of the Apennines.





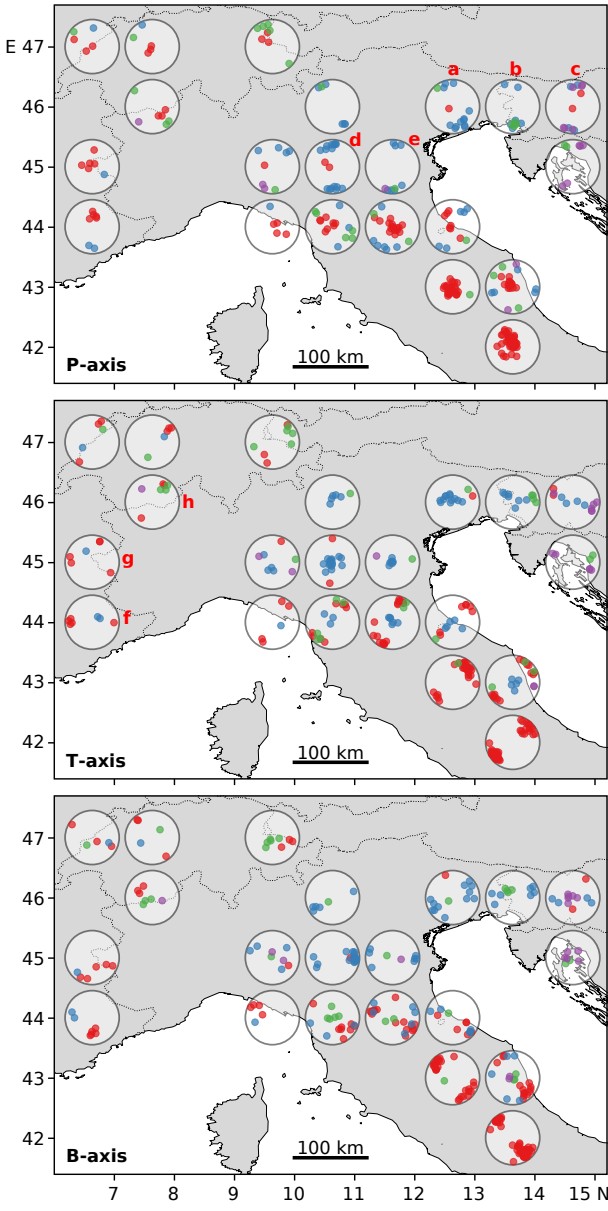

**Figure 10.** Regional distribution of P-, T- and B-axes of focal mechanisms presented in Fig. 9. Colors correspond to mechanism classes as shown in Fig. 9. Only areas with more than five events in a latitude-longitude-grid of 1°x1° are shown. a to h mark features which are discussed in the text.

The distribution of P- and T-axes across the Alps points out both, local and regional heterogeneities (Fig. 10). While in general compressional regimes are dominant in the SE Alps and the central Alps, extensional stresses are observed within the western Alps. In the SE Alps, across a distance of 200 km, a rotation of the P-axes from NW-SE in the western part to NNE-SSW in the eastern part is observed (Fig. 10a, features a-c). At the southern margin of the Alps, we observe predominantly





thrust mechanisms with NW-SE oriented P-axes in the central Alps, close to Lake Garda, to NNE-SSW oriented P-axes further

east, close to Vicenza (Fig. 10a, features d and e). Despite the small number of moment tensors, we can also infer a rotation of the T-axes in the western Alps. The T-axes of the focal mechanisms are oriented roughly perpendicular to the bending of the Alpine arc (Fig. 10b, features f-h). This observation is in accordance with Mathey et al. (2020), who recently studied the stress field of the western Alps based on an new extensive dataset of P-wave first motion polarity focal mechanisms.

In addition to the direct interpretation of P- and T- axes of the MT solutions, we apply a stress inversion approach based on

the minimization of the seismic energy released on unfavourably oriented faults (Cesca et al., 2016) in volumes of high seismic activity (Supplement Fig. S2). Stress inversion results provide the orientations of the most compressive ($\sigma_1$), the intermediate ($\sigma_2$) and the least compressive principle stresses ($\sigma_3$), and the relative stress magnitude R=($\sigma_1$-$\sigma_2$)/($\sigma_1$-$\sigma_3$). A homogeneous stress field is assumed within the selected rock volume and time period for each subregion. Our stress inversion results confirm dominating compression from the central to the eastern Alps with sub-horizontal $\sigma_1$ orientation and extension in the western

Alps with a sub-vertical $\sigma_1$ orientation. Despite increased uncertainties due to the relatively low number of available MT solutions, we observe the same rotation of $\sigma_1$ from the central Alps to the eastern Alps from NNW-SSE close to Lake Garda and to Friuli to NNE-SSW in the northern Dinarides. The transition from dominant thrust faulting in the south-eastern Alps close to Friuli to the strike slip events to the east and in the northern Dinarides is mapped by the change from a sub-vertical to an almost horizontal $\sigma_3$ direction.

**4.2 Distribution of centroid depths**

In Fig. 11, we display the depth distribution of mechanisms depicted in Fig. 9 sorted by faulting type. In the left panels, the focal mechanisms derived from the afore mentioned bulletins are shown, while our own centroid solutions are provided in the right panels. While the depth in the catalogs may be partly fixed during the inversion, the centroid depth is determined during the MT inversion in our approach. Uncertainties of our solutions are mostly in the range of 1 to 3 km. Many events within the

Alpine mountain range are shallower than 10 km (Fig. 11d-f). While we obtained depths below 5 km for all normal faulting events in the NW Alps, depths of up to 15 km are observed for thrust faulting events in the SE Alps.

Centroid depths in the Apennines can be significantly larger. Thrust and normal faulting events are roughly separated into two NW-SE running bands, with more thrust events in the NE and more normal faulting events in the SW part. Normal faulting and strike-slip events occur predominantly in shallower depths (<20 km) than the thrust faulting events with depths of often

30-50 km.

The observed depth ranges are in accordance with the maximum depths in the long-term seismic catalogs of gCMT, INGV and GEOFON, including >50,000 earthquakes with Ml>2.0 (Fig. 12c). These catalogs show that seismicity is shallow across most of the Alps, while maximum depths of above 60 km are observed in the Apennines.



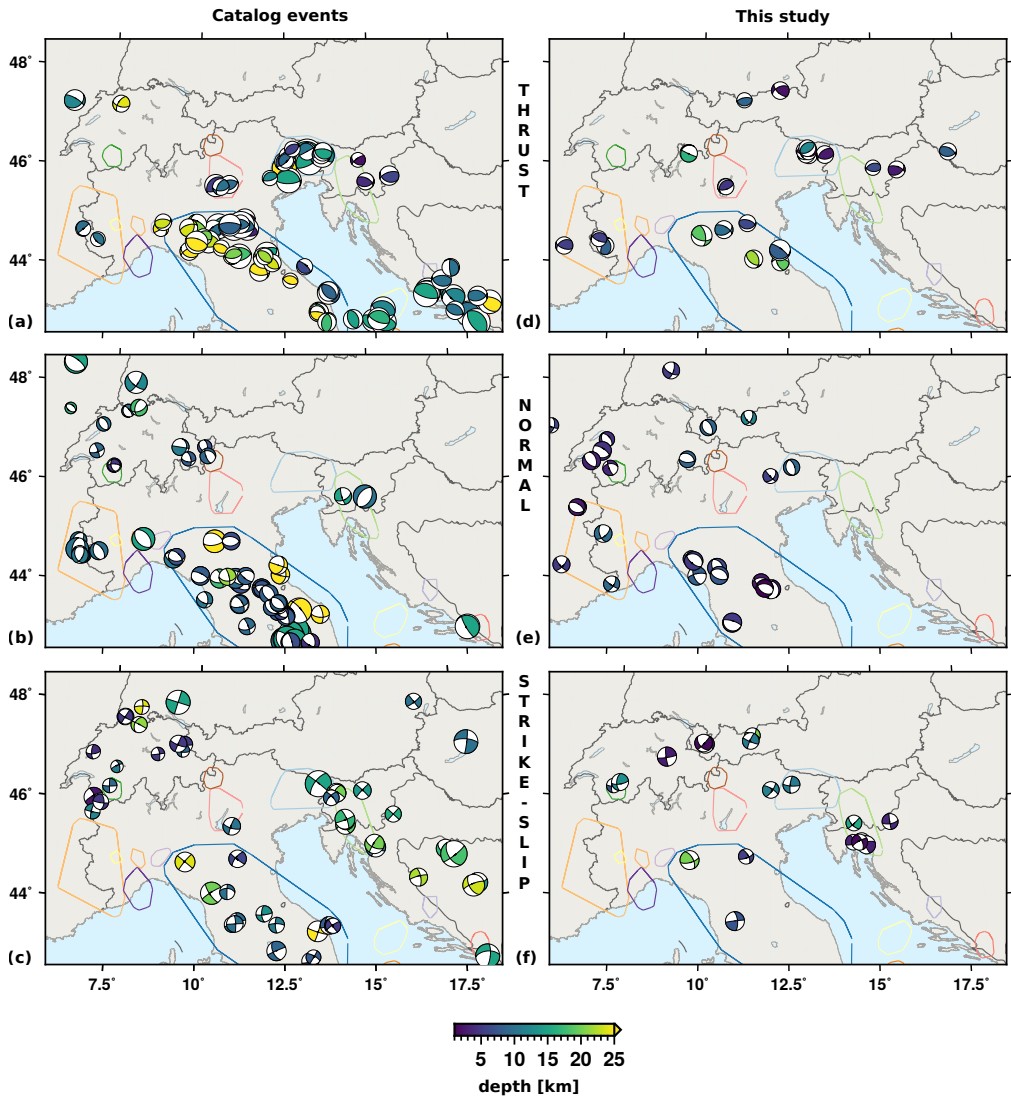

**Figure 11.** (a)-(c) Catalog depths of earthquakes shown in Fig. 9. (Depths might be fixed in some catalogs.) (d)-(e) Centroid depths of MT solutions in this study. Earthquakes are sorted according to their mechanisms: (top) thrust faulting events, (center) normal faulting events and (bottom) strike-slip events. The outlines of spatial clusters of increased seismic activity from Fig. 12a are indicated for comparison.





### 4.3 Seismicity and strain

Since only few focal mechanisms are available for large parts of the Alps, we rely on recent seismicity, historic large earthquakes and GNSS data to further characterize the tectonic activity. To emphasize areas of high seismic activity (Fig. 12a) we cluster the earthquakes in the seismicity catalogs by INGV, GEOFON and gCMT according to their epicentral locations using the DBSCAN clustering algorithm (Ester et al., 1996) implemented in the python package *scikit-learn* (Pedregosa et al., 2011). The merged catalogs comprise a total of more than 50,000 earthquakes with Ml>2.0 (1983-2017). Additionally, we report maximum hypocenter depths and cumulative seismic moments across the study area (Fig. 12c and d).

In general, the seismicity along the southern margin of the Alpine mountains is higher than along its northern counterpart. Apart from the large cluster of seismicity in the Apennines, we identify five dominant clusters and several smaller ones mainly located at the margins of the Alps (Fig. 12a). Following the Alpine arc from west to east, the first cluster is located in the western Alps at the French-Italian border, two smaller clusters are found in the region around Lake Garda in the S central Alps and north of it. Two more clusters of high seismicity are situated in the SE Alps in the border region between Italy (Friuli) and Slovenia, and in the northern Dinarides. In contrast, the eastern Alps north of the Periadriatic line and the area between the high seismicity regions in the western Alps and the central Alps have particularly low seismicity rates (Fig. 12a).

The recent activity in the seismicity clusters described above can be traced back to historical times. Fig. 12b shows historic earthquakes from the European Archive of Historical Earthquake Data 1000-1899 (AHEAD, Locati et al. (2014); Rovida and Locati (2015)) based on the SHARE European Earthquake Catalogue (SHEEC, Stucchi et al. (2013)) and the ISC-GEM catalog (Storchak et al., 2013, 2015; Bondár et al., 2015; Di Giacomo et al., 2015, 2018) with Mw>5.5. Large historic events are reported for the SE Alps (Slovenia 1511 and Carinthia 1348, both Mw >6.8) and close to Lake Garda in the S central Alps (Verona 1117, Mw 6.7). Large historic earthquakes with magnitude estimates between Mw 5 and 6 are also reported in the western Alps and the Dinarides. While at least three large earthquakes occurred in the eastern part of Switzerland in historic times, seismicity in this region appears to be relatively low in recent years (Fig. 12a and b). The cluster of high seismicity in the SE Alps (Fig. 12a) is located close to the epicenter of the 1976 Friuli Earthquake (Mb 6.0, Pondrelli et al. (2001)). The observed E-W striking thrust events map the regional dominant stress field (Fig. 9), evolving from the underthrusting of the Friuli Plain beneath the Alps (e.g. Cipar (1980)). Focal mechanism solutions of the 1976 mainshock and aftershocks show similar thrust focal mechanisms striking E-W to NE-SW, partly with a small strike-slip component, and are associated to the complex Periadriatic overthrust system (e.g. Cipar (1980), Pondrelli et al. (2001), Pondrelli et al. (2006), Slejko (2018), Poli and Zanferrari (2018), Bressan et al. (1998)). East of Friuli, in the seismic cluster of the northern Dinarides (Slovenia and Croatia), we observe dominant strike-slip faulting mechanisms In addition, only few tenth of kilometers to the west of the Friuli area, close to Forni di Sopra, Italy, we observe strike-slip faulting (Fig. 9). Anselmi et al. (2011) report the occurrence of both thrust and strike-slip faulting for this area, mostly in agreement with an E-W to ENE-WSW minimum horizontal stress reported by Montone et al. (2004). Within the seismicity cluster close to Lake Garda, Italy, we observe NE-SW striking thrust faulting. These mechanisms are typical for earthquakes located in the Giudicarie region close to the Ballino-Garda fault (Viganò et al., 2008).





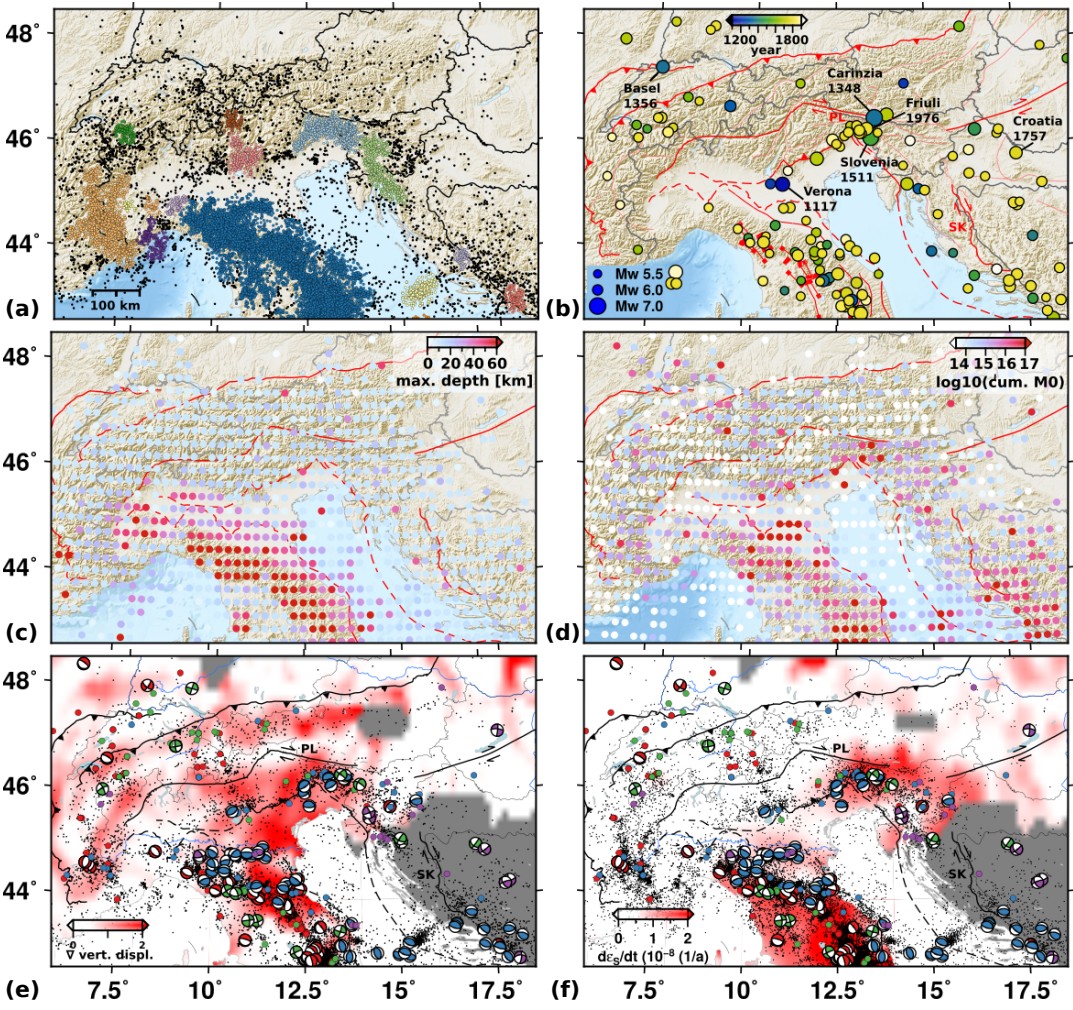

**Figure 12.** Characteristics of recent and historic seismicity and strain from GNSS data. (a) Seismic activity between 1978-2017, Ml >2.0, from GEOFON, INGV and gCMT catalogs, colored according to epicentral clusters of seismicity. (b) Historic earthquakes with Mw >5.5 from the European Archive of Historical Earthquake Data 1000-1899 (AHEAD) (Locati et al., 2014; Rovida and Locati, 2015; Stucchi et al., 2013) and ISC-GEM catalog (1906-2016) (Storchak et al., 2013, 2015; Bondár et al., 2015; Di Giacomo et al., 2015, 2018). (c, d) Maximum event depth and the cumulative seismic moment on a grid with a spacing of 0.25x0.25° lat. and lon. (e, f) Absolute value of the spatial gradient of the relative uplift rate as a proxy for vertical strain rates and observed GNSS shear strain rate (max. shear strain). Both obtained from GNSS data of the EUREF WG on European Dense Velocities (Brockmann et al., 2019). Grey dots indicate earthquakes without MTs, as in (a). Events with MT solutions are color-coded as in Fig. 9. For simplicity, focal mechanisms are only plotted for representative events with Mw≥4.0. Exposed and subsurface faults (solid and dashed lines) simplified from Schmid et al. (2004, 2008); Handy et al. (2010, 2015) and Patacca et al. (2008). 'PL' marks the Periadriatic line, 'SK' the Split-Karlovac Fault. Topographic data from SRTM-3 (Farr et al., 2007) and ETOPO1 (Amante and Eakins, 2009) datasets).





In the last decades, highest cumulative seismic moments (Fig. 12d) are observed within the seismicity clusters of Fig. 12a.
Earthquakes with magnitudes above 5.0 are rarely observed in the Alps and cluster in the Apennines and the Friuli area. Within
the time span of >30 years, earthquakes with $4<Ml<5$ are observed across a wider part of the western and southern Alps. In
contrast, in large areas of the central to NE Alps, magnitudes rarely exceed Ml 3.5. Therefore, only few focal mechanisms are
available in this area.

Fig. 12e and f present the spatial gradient of the uplift rates and the horizontal strain rates, computed from the GNSS data
of the EUREF WG on European Dense Velocities (http://pnac.swisstopo.admin.ch/divers/dens_vel/index.html, (Brockmann
et al., 2019)). Please refer to the supplement for additional methodological information. Following Keiding et al. (2015), we
use the spatial derivative of the uplift rate as a proxy of vertical strain rates (Fig. 12e).

Within the Alpine mountain range, the GNSS data shows a consistent uplift relative to the surrounding areas (Supplement
Fig. S3). Fig. 12e and f emphasize the relation between recent seismic activity and both, high spatial gradients of the uplift rate
(e) and the shear strain rate (f). Largest gradients of the uplift rate and high shear strain rates are observed in the SE Alps and the
Apennines where seismicity is highest. While there is no significant shear strain in the western and central Alps, we depict two
subparallel bands of moderate spatial gradients of the uplift rate running roughly along the northern and the southern margin of
the Alps. These two elongated regions also have higher seismicity rates compared to the central Alpine belt. We observe that
thrust faulting events cluster at the SE margin of the Alps. South-east of this area, the uplift gradients are low, while increased
shear strain rates agree with our dominant strike-slip mechanisms. In the SW Alps, the largest events cluster in the transition
area between relative uplift and subsidence, indicated by the band of increased spatial gradient of the uplift in Fig. 12e. Normal
faulting events are dominant. Intraplate shear strain rates are relatively low in the entire western and central Alps. The seismic
activity along the northern margin of the Alps is in agreement with the increased gradient of uplift in this area. However, the
occurrence of strike-slip faulting earthquakes described in this study can hardly be explained by vertical strain, especially for
favourably oriented faults. However, pre-existing faults may be unfavorably oriented, the stress field may be heterogeneous
and local anomalies may not be resolved by the sparse GNSS network. Rather low seismicity is further observed in the eastern
Po plane, where high spatial gradients of the uplift rate are observed. The high absolute gradient here can be attributed to the
relative subsidence of the sediments in the Po plane (see supplement Fig. S3), but not to a tectonic uplift which would likely
be accompanied by seismic activity.





# 5 Conclusions

Centroid moment tensor inversion provides insight into faulting mechanisms of earthquakes and related tectonic processes. In this study, we used the AlpArray seismic network to analyze the mechanisms of earthquakes occurring from 2016 to end of 2019. Thanks to the flexible inversion tool *Grond*, we were able to test different inversion set-ups and to evaluate the

results with respect to their uncertainties and parameter trade-offs. For subsets of events, we tested various frequency bands, distance ranges and different input data types comprising time domain full waveforms, frequency domain amplitude spectra, time domain cross-correlation fitting, waveform envelopes and combinations of these. We evaluate the influence of azimuthal gaps and study both the occurrence and the meaningfulness of non-DC components. We propose to perform similar tests prior to MT inversions for other study areas, when magnitudes are small, the number of stations is limited or other factors might

hinder straight-forward inversions. In case of our study area, for most earthquakes with magnitudes larger Mw 3.3, we find that a combination of time domain full waveforms and frequency domain amplitude spectra in a frequency band of 0.02-0.07 Hz is most suitable. We observe that even in cases where non-DC components seem well resolved, these components should be carefully evaluated by forward modelling and comparing the waveforms of solutions from pure-DC, deviatoric and full inversions. In case of small events, we observe that large gaps in the azimuthal station distribution can hinder successful

inversions or may lead to biased results. In contrast, under favorable geometric conditions, when strike direction and tensional as well as compressional quadrants are covered, MT inversions are possible even for small events using only few stations.

Relying on the results of the methodological tests, we performed deviatoric MT inversions for events with Mw>3.0. We present 75 solutions with reasonably low uncertainties for earthquakes with Mw>3.1 occurring between 2016 and 2019 and compare these to historic earthquakes, recent seismicity and published focal mechanisms. Our moment tensor results indi-

cate that while the Alps represent a rather heterogeneous study area, the region is characterised by compartments of different tectonic movement in close proximity. We discussed these different tectonic areas systematically, including observations of seismicity, faulting mechanisms and GNSS deformation data. Based on a clustering of epicenters, we identify five main seismically active subregions, namely the western Alps, the region around Lake Garda, the SE Alps, the northern Dinarides and the Apennines. These regions are mostly located in the proximity of the southern margin of the Alps, where significant vertical

or horizontal strain rates are reported. In contrast, seismicity is particularly low in the eastern Alps and in parts of the central Alps. Both the depths inferred from our moment tensor inversions as well as the depths in seismic catalogs indicate that the seismic activity in the Alpine mountain ranges is predominantly shallow. Significantly deeper earthquakes are observed in the Apennines. Maximum observed magnitudes coincide with regions of increased seismicity but rarely exceed Mw 5.0. We applied a clustering based on the Kagan angle to identify groups of similar events. Typical ENE-WSW to E-W striking thrust

faulting is observed in the Friuli area in the SE Alps. Strike slip faulting with similarly oriented P-axes are observed parallel to the northern margin of the central Alps and in the northern Dinarides. In contrast, NW-SE striking normal faulting events with NE-SW oriented T-axes are observed in the NW Alps with similar strike directions but at shallower depths than the dominant normal faulting events in the Apennines. Oblique to the bending of the Alpine mountain range, the T-axes of normal faulting earthquakes rotate from NW-SE in the NW Alps to about E-W in the SW Alps. While in the Apennines mechanisms types are





separated along the mountain range by depth and location, within the Alps we observe a separation of compressional stresses
in the SE as well as in the central Alps and a tensional regime in the western Alps.

While we are able to obtain stable and reliable MT solutions for most earthquakes with Mw $\leq$ 3.3 that occurred during the
installation period of the AASN between 2016 and 2019, we struggled to invert smaller events with Mw 3.1-3.3. On one hand,
the spacing of the AASN of 60 km is not sufficiently dense for smaller earthquakes and on the other hand, topographic effects,
low SNRs, as well as complex subsurface structures complicated the inversions, resulting in a success rate of one third for the
magnitude range Mw 3.1-3.3. We strongly recommend a data quality assessment before performing any inversion. In future,
we plan to model topographic effects and include recent 1-D and 3-D velocity models developed in the course of the AlpArray
research initiative.



*Code and data availability.* The moment tensor inversions were performed using the free and open-source inversion tool grond (Heimann et al., 2018). Figures were plotted using pyrocko (Heimann et al., 2019) and GMT (Wessel et al., 2013).

The topographic data for the maps was taken from the SRTM-3 (Farr et al., 2007) and ETOPO1 (Amante and Eakins, 2009) (NOAA National Geophysical Data Center. 2009: ETOPO1 1 Arc-Minute Global Relief Model. NOAA National Centers for Environmental Information. Accessed [December 2020].)


Seismic catalogs: Earthquake information and focal mechanisms are provided by the below mentioned institutes:

Swiss Seismological Service (SED): (Deichmann et al., 2004; Baer et al., 2005; Deichmann et al., 2006; Baer et al., 2007; Deichmann et al., 2008, 2009, 2010, 2011, 2012; Diehl et al., 2013, 2014, 2015, 2018; Diehl, 2020)

Slovenian Environment Agency (ARSO): Ministrstvo za okolje in prostor Agencija RS za okolje (2018), Ministrstvo za okolje in prostor

Agencija RS za okolje (2019), Ministrstvo za okolje in prostor Agencija RS za okolje (2020)

INGV (Italy): http://terremoti.ingv.it/en (last access 12/2020)

GEOFON (Germany): Event locations were obtained from the GEOFON program of the GFZ German Research Centre for Geosciences using data from the GEVN partner networks. https://geofon.gfz-potsdam.de/eqinfo/list.php (last access 12/2020)

EM-RCMT (European-Mediterranean Regional Centroid-Moment Tensors): http://rcmt2.bo.ingv.it/ (last access 12/2020), Pondrelli (2002)

SISMOAZUR (France): http://sismoazur.oca.eu/focal_mechanism_emsc (last access 12/2020), - Provide MTs obtained using FMNEAR (Delouis, 2014)

gCMT (Lamont-Doherty Earth Observatory of Columbia University, USA): https://www.globalcmt.org/ (last access 12/2020), Dziewonski et al. (1981); Ekström et al. (2012)

Permanent seismic networks: The permanent stations of the AlpArray are part of existing European regional networks (RD (RESIF, 2018) , GU (University Of Genova, 1967), CZ (Institute Of Geophysics, A. O. S. O. T. C. R., 1973), ST (Geological Survey-Provincia Autonoma Di Trento, 1981), G (Institut De Physique Du Globe De Paris (IPGP), & Ecole Et Observatoire Des Sciences De La Terre De Strasbourg (EOST), 1982), CH (Swiss Seismological Service (SED) At ETH Zurich, 1983), OE (ZAMG-Zentralanstalt Für Meterologie Und Geodynamik, 1987), MN (MedNet Project Partner Institutions, 1990), HU (Kövesligethy Radó Seismological Observatory (Geodetic And Geophysical Institute,

Research Centre For Astronomy And Earth Sciences, Hungarian Academy Of Sciences (MTA CSFK GGI KRSZO)), 1992), GE (GEOFON Data Centre, 1993), RF (University Of Trieste, 1993), FR (RESIF, 1995), IV (INGV Seismological Data Centre, 2006), BW (Department Of Earth And Environmental Sciences, Geophysical Observatory, University Of Munchen, 2001), SX (Leipzig University, 2001), NI (OGS (Istituto Nazionale Di Oceanografia E Di Geofisica Sperimentale) And University Of Trieste, 2002), TH (Jena, F.S.U., 2009), OX (OGS (Istituto Nazionale Di Oceanografia E Di Geofisica Sperimentale), 2016)).

**Team list**

The complete member list of the AlpArray Working Group can be found at http://www.alparray.ethz.ch.

*Author contributions.* G.P., S.C. and T.D. conceptualized the study. G.P. performed the moment tensor inversions, conceptualized and developed the methodological tests, prepared the figures and wrote the original draft of the manuscript. S.C., T.D., J.K. and T.P. developed





the general project idea and acquired the funding. S.C., S.H., D.K. and T.D. supervised the study. S.H. et al. developed the moment tensor

inversion and the software tools used in this study. S.H., S.C., T.D. and P.N. contributed to methodological developments. S.H. and P.N. helped to visualize the results. T.D and T.P. performed stress field inversions. G.P, S.C. and T.D. validated the results. The AlpArray Working Group provided access to the seismic data of the temporary network and additional background information. All Co-authors reviewed and edited the manuscript.

*Competing interests.*   The authors declare that they have no conflict of interest.

*Acknowledgements.*   GMP is funded by DFG project 'From Top to Bottom - Seismicity, Motion Patterns and Stress Distribution in the Alpine Crust' (Project Number 362440331), a subproject of 'SPP 2017: Mountain Building Processes in 4D' (Project Number 313806092). PN is currently funded by the BMBF (German Federal Ministry of Education and Research) project SECURE (grant agreement No. 03G0872A).



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
