# Peer review of "Regional centroid MT inversion of small to moderate earthquakes in the Alps using the dense AlpArray seismic network: challenges and seismotectonic insights"

_Solid Earth, 2021_

## Referee Comment (RC2)

[referee-annotated manuscript omitted]

---

## Author Response (AR1)

Dear Reviewers, dear Editor,

we are thankful for the helpful and thorough reviews. The suggestions and questions by the three reviewers helped us to improve our manuscript.

We tried to follow the suggestions by all reviewers and answer below to the remarks of each reviewer separately. We provide the answers to all reviewers in the same document to facilitate an overview of all requests and suggestions for changes in the manuscript.
Additionally, we include an annotated version of the manuscript, in which changes are marked in red.

Besides smaller improvements and corrections, we followed the suggestions by reviewer 2 and 3 and reworked and restructured the discussion. We moved the subsection on the CMT depths as well as the part on the clustering based on focal mechanisms into the result section. Furthermore, we followed the suggestions to improved the discussion by including more comparisons to existing literature. Additionally, we modified the conclusions to better highlight the most important findings of our study.

We improved the introductory Figure 1, which now provides more information on locations and faults which are mentioned in the text.

Finally, we now follow Alpine naming conventions which usually subdivide the Alps into the Western, Central, Eastern and Southern Alps. Consequently, we now refer to the eastern Southern Alps, which we previously described geographically as the South-Eastern Alps.

With kind regards on behalf of the authors,

Gesa Petersen

**Reviewer Comment 1**

This study uses an unprecedented dataset from dense seismic deployments (AlpArray) in the wider Alpine area and a versatile moment tensor inversion tool in order to show the viability of full moment tensor inversion for small regional events, and to produce a large set of solutions that bring forward our understanding of regional deformation. In my opinion, this represents a solid and useful contribution and should appear in Solid Earth. Besides the value for Alpine seismotectonics, strength of this manuscript is the extensive testing to explore processing parameters. This provides important hints for setup and parameter choices (filter bands, etc.) in other moment tensor initiatives directed at small events. The authors also simulate less dense networks than AlpArray, which will be the case for most applications. Also, the work explicitly addresses the appearance and significance of non-DC components in moment tensor sources, without prejudice as weather these components are expressions of a plausible source process or artefacts from modelling. Final moment tensor solutions are consistent within this dataset as well as in comparison with previous solutions, validating the inversion procedure and helping in the delimitation of seismotectonic domains in the region. Beyond analysis of the present moment tensor data set, the authors extend the discussion to previous seismicity, to provide a general seismotectonic summary for the wider Alpine area. This manuscript contains a lot of information and is suited to arouse curiosity and further questions, making it well suited for Solid Earth as an interactive journal. To start, here I propose points for further thinking or minor revision of this manuscript:

- **Q**: 1) Effects of station coverage: In section 2.2.4, tests show how the reduction of azimuthal coverage affects moment tensor estimates. However, formally, one single three-component station is sufficient to resolve a DC mechanism (if the Earth model and everything else is ok, in practise this should be avoided). Did the authors try the comparison using DC-constraint in inversion or comparing DC-components of full moment tensors? Is the DC information more stable than full moment tensors if the azimuths become narrow?

  *A: Section 2.2.4 discusses the effects of the station coverage assuming a deviatoric (not full) moment tensor. The reviewer is right to point out that we did not discuss the resolution of the pure-DC component. While we do not run a constrained DC inversion, we assess the stability of the DC component, as obtained from the DC+CLVD decomposition of the deviatoric MT. We have now included a few sentences on the results of this evaluation and mention in the text that – in theory – the DC can be resolved relying on a single station.*

  *→ p. 18 in the annotated manuscript:*

  In theory, the DC components of a moment tensor can be resolved from a single station using 3-component data (Dufumier and Cara, 1995). However, such an analysis requires a high data quality and exact knowledge about velocity structures and path effects. In practice, single-station approaches are mostly avoided as they often result in unstable solutions (Dufumier and Cara, 1995).

  Dufumier, H. and Cara, M.: On the limits of linear moment tensor inversion of surface wave spectra, pure and applied geophysics, 145, 235–257, 1995.

  *→ p. 18 in the annotated manuscript:*

  Fig.8 shows the fuzzy MTs (right panels) for decreasing azimuthal coverage of seismic stations (left panels) for three exemplary events. In the case of the largest event (Mw 4.1), the solution is very stable when seismic stations cover at least an azimuthal range of 90°. In the case of an even smaller coverage, the mechanism rotates slightly depending on the azimuthal direction of the remaining stations. In the case of the Mw 3.9 event, the uncertainties of the solutions increase with decreasing station coverage. Two examples in which the inversions were done with stations covering only an azimuthal range of 45° show significant differences between the resulting focal mechanisms. When only considering the fuzziness of the two focal mechanism plots, both ensembles of solutions seem to be well resolved and stable. This indicates that the amount and variability of input data is not sufficient to resolve the MT unambiguously. Furthermore, we observe a clear trend of increasing non-DC components with decreasing azimuthal coverage. We find a non-DC component below 10 % for a coverage of 180°, but 40 % for the smallest tested coverage for the Mw~3.9 event.

  For the smallest earthquake (Mw~3.5), the resulting MT solutions vary even more. In the case of the inversions with a station coverage of 90°, the variability among the ensembles of solutions is high and depends on the location of the 90° quadrant covered with stations. When only considering the dominant DC components of the deviatoric moment tensors, we observe the same general correlation between coverage and resolution. It is worth noticing that even with a small number of stations covering a small azimuthal range, it is possible to resolve a MT under favorable geometrical conditions. When stations are located in strike direction and cover both tensional and compressional quadrants, they may resolve the MT correctly even when covering only 45° (Fig. 8, Mw 3.5 event, 5th row and last row). [...]

- **Q**: 2) Geodynamic interpretation: Earthquake and GNSS data agree on the characteristics of deformation in the SE-Alps and Apennines, but the western and central Alps appear more enigmatic. The results from moment tensor inversion (extensional stresses, with T- axes oriented rather perpendicular to the bending of the arc, and shallow depths) and GNSS data (absence of convergence, or any other significant regional strain) apparently provide support for previous models that attribute seismicity to gravity and buoyancy forces (e.g. Delacou et al., 2004, Geophys. J. Int 158, 753–774). The authors point out the coincidence between faulting in the Western Alps and uplift, but probably could add more thoughts whether the results could be interpreted in an active (tectonic) or passive (erosion or deglaciation) framework.

*Thanks for pointing at this interesting observation. We extended the discussion based on this suggestions and on similar ones by the other reviewers, including the suggested references (and others).*

*→ p. 30 in the annotated manuscript:*

While there is no significant shear strain in the Western and Central Alps, we depict two subparallel bands of moderate spatial gradients of the uplift rate running roughly along the northern and the southern margin of the Alps. These two  bands result from the overall relative uplift of the Alps and also have higher seismicity rates compared to the central Alpine belt. In the SW Alps, the largest events cluster in the transition area between relative uplift and subsidence, indicated by the band of increased spatial gradient of the uplift in Fig. 12e. Normal faulting events are dominant. Intraplate shear strain rates are relatively low in the entire Western and Central Alps.

The Adriatic plate, which is the upper plate in the Alpine subduction zone, rotates counter-clockwise relative to Europe around an Euler pole located in the western Po plain or Western Alps (D'Agostino et al., 2008; Weber et al., 2010; Le Breton et al., 2017). The rotation results in varying convergence rates across the Alps. Le Breton et al. (2017); Le Breton et al. (2021) infer a rotation of about 5.25° during the last 20 Ma resulting in convergence rates ranging from 5.5 mmyr$^{-1}$ in the NW Adria (Western Alps) to 7.5 mmyr$^{-1}$ in the NE Adria (eastern Southern Alps) (Fig. 1). In comparison, kinematic reconstructions of Van Hinsbergen et al. (2020) involve less convergence but a higher rotation of Adria relative to Europe (10°), leading to 2.5 mmyr$^{-1}$ convergence in the NW Adria to 6.25 mmyr$^{-1}$ in the NE Adria. Recent GPS data indicates little to no horizontal movement in the Western Alps but more than 2 mmyr$^{-1}$ NNW-ward movement of Adria in the eastern Southern Alps (see Fig. S3 in the Supplement), which is in agreement with increased seismicity rates. The Western Alps are closer to the location of the Euler pole of the rotation of the Adriatic plate, therefore convergence rates are lower. Recent GPS measurements and the computed horizontal strain rates even indicate the absence of convergence (D'Agostino et al., 2008, , and Fig. S3 in the Supplement). Therefore, the uplift pattern of the Western and Central Alps (Fig. 12e and Fig. S3 in the Supplement) as well as the seismicity clusters in the W Alps need to be attributed to other mechanisms. Sternai et al. (2019) propose that isostatic adjustment to deglaciation and erosion, and mantle-related processes such as slab detachment or asthenospheric upwelling may jointly explain the observed uplift pattern. The assumption of a stress/strain field which is not dominantly effected by the convergence of Europe and Africa was also proposed by Delacou et al. (2004) based on focal mechanisms and stress inversion in the Western Alps. Our moment tensor solutions indicating normal and strike-slip faulting, as well as the P and T axes match these observations from GNSS data.

- 3) Technical details:

  - Q: How are crust2.0 models assigned: according to source side structure, receiver side structure, or both (given that QSEIS could handle two different crustal structures)?

  - *A: We choose the velocity model according to source site structure. We added this information in the text. We also tested the inversion using various crustal models for the station sites, without a significant improvement. This is possibly expected, when fitting mostly surface waves.*

    *→ p. 6 in the annotated manuscript:*
    Precalculated Green's function data bases are used for rapidly computing synthetic data (Heimann et al. 2019). In our case, we used regional velocity profiles from the CRUST2.0 Earth model database (see http:/igppweb.ucsd.edu/~gabi/crust2.html, last access June 2020, and Bassin et al. 2000), which we choose according to the earthquake epicenter location.

  - **Q**: Small time shifts are allowed in time domain full waveform inversion and regulated with a penalty function, which may be to some extend a way to clear inconsistencies between time domain, envelope, cross-correlation and spectral inversion. Question: how small are these time shifts, and how penalizing the penalty function? To answer this question, maybe also a more general statement about the combination of data misfit is necessary.

    *A: The reviewer is right, this information was missing in the manuscript. The maximum allowed time shift was set according to be well below a quarter of the dominant wave length, i.e. about 4s for the surface wave inversions, using a band-pass filter between 0.02-0.07 Hz and finding dominant frequencies in the order of 0.02-0.05 Hz. The maximum penalty was empirically set to 0.05. Penalties for time shifts are applied on single-station-component misfit level.*
    *We included a small statement to inform the reader about the weighting of the input data types and the misfits.*

    *→ p. 5 in the annotated manuscript:*

    The objective function is set up in a flexible way to combine different input data types as a weighted sum. In our study, we use combinations of time domain full waveforms, time domain cross-correlations and frequency domain amplitude spectra as an input for the inversion. Following the studies of Zahradnik et al. (2018) and Dahal et al. (2020), we implemented envelopes of time domain waveforms. The misfits of the different input data are combined using an L1 or L2 norm. We assign the same weighting to each input data type. The misfit values of the single stations within one input data type group are weighted to account for different epicentral distances Heimann et al. 2011. Without applying such a weighting, summed misfits are always dominated by the closest stations which have the highest amplitudes. In the case of using time domain full waveform or cross-correlation fitting procedures, we allow for small time shifts to compensate for errors in the velocity models. To avoid the mis-matching of phases, these time shifts were set to be well below a quarter of the dominant wavelength. Time shifts are regulated by a penalty function with an empirically chosen maximum of 0.05.

  - **Q**: The authors identify as principle problem of first motion mechanism the fact that they are only representative for the rupture onset and cannot represent complex rupture

processes. Two comments: Point source MT cannot represent general complex sources either; and second: there may be more relevant problems for first motion mechanisms in practise, like for example the instability of take-off angles for shallow events.

***We agree with the reviewer and followed the suggestion. We removed the statement on the complex sources and instead added a statement pointing out that errors can be introduced due to the instability of take-off angles of shallow events.***

*→ p. 4 in the annotated manuscript:*

In contrast, many of the previous studies focus on specific regions or seismic sequences within the Alps, therefore not providing a broad overview. Furthermore, many of these studies relied on first motion polarities. First-motion based approaches can be used even for small earthquakes when no surface wave energy is observed. However, the obtained mechanism is only representative for the very first moment of the fracturing process.  This might introduce discrepancies when comparing first motion solutions to MT solutions (Scott and Kanamori 1985, Guilhem et al. 2014). The instability of take-off angles of shallow earthquakes may introduce significant errors in the polarity readings (Hardebeck and Shearer 2002). Additionally, first motion solutions of small earthquakes are often only based on few polarities, which makes it difficult to assess uncertainties.

**Reviewer Comment 2**

This paper describes the computation of focal mechanism and moment tensors for the seismicity recorded by the AlpArray temporary network. The Alpine region has often mainly a low magnitude seismicity for which the authors were able to obtain good quality focal mechanisms, really important because usually lacking.
The paper is Interesting, mostly well written and rich of methodological descriptions, tests and information. However, the discussion part is too long, sometime redundant and after several elaborations, it is lacking of new conclusions. Indeed in the Conclusion paragraph quite nothing is said about seismotectonic reasoning.

I consider the paper needs really minor changes in the first part (methodologies and computations, see comments in the pdf file), while it would benefit from a more synthetic version of the Discussion part (see again comments in the pdf file).

Some smaller comments follow here:

- Q: In the abstract is not defined the time window of described seismicity.

  **A: *Thanks for pointing this out. We have included the information in the Abstract.***

- Q: Indeed, western Alps in general cannot be considered as an high seismicity region.

  **A: *This is correct, we adjusted the text to make sure that we talk of high seismicity only compared to the rest of the Alps and within a long time period.***

- Q: All reference format in the text should be corrected.

*A: Thanks a lot for pointing out that we did not use the SE reference format. We have now corrected this.*

I appreciated a lot all tests performed to asses when it is really useful to perform a super complete analysis (full MT inversion), being aware that otherwise the results are related to ambiguities and uncertainties due to the quality of seismograms in term of amount, azimuthal distribution and SNR.

- Q: In the discussion some descriptions are redundant or not meaningful. For instance, describe together the extensional focal mechanisms of the western Alps and those in the Apennines, saying also they have a similar strike does really add an useful information? In my opinion, no. The authors may describe them separately to decrease any possible confusion.

  *A: We followed the suggestions by reviewer 2 and 3 and restructured and improved the discussion. To avoid confusion, we follow the suggestion to describe the normal faulting events in the Apennines and Western Alps separately.*

  *→ p. 26 in the annotated manuscript:*

  Normal faulting, with a similar orientation as in the NW Alps, is dominant along the central arc of the Apennines. The normal faulting events of the NW Alps are located within the strike direction of those in the Apennines. However, despite the similarity of mechanisms, the depths of the events in the NW Alps are significantly shallower (Fig. 11).

  Along the Apennines, thrust faulting is dominant at the northern arc, while normal faulting earthquakes are dominant south-west of the ridge of the Apennines. The NW-SE orientations of the T axes of the normal faulting events are perpendicular to the elongation of the mountain belt as also described by Pondrelli et al. (2006). The vertical $\sigma_1$ direction and the NE-SW oriented, horizontal $\sigma_3$ direction confirm an extensional stress regime (Supplement Fig. S2). In contrast, a compressional regime is observed along the NE arc of the Apennines with P axes of the thrust faulting events oriented NW-SE to NE-SW (Fig. 11).

- Q: Please, check the description of Supplementary material. In the first part some sentences are not comprehensible.

  *A: We are sorry that the descriptions in the supplement were not well formulated and corrected them.*

**Comments by reviewer 2 in the pdf file:**

*A: We corrected all typos indicated in the pdf. Here we respond to the questions that were additionally raised in the comments in the manuscript by reviewer 2.*

- Q: line 26: upthrusted – do you mean it is the upper plate in a subduction system?

  A: *This sentence was clearly not well formulated, we corrected it:*

  *→ p. 2 in the annotated manuscript:*

  Geological studies show that the Adriatic plate was upthrusted is the upper plate in the subduction of the Alpine Tethys in the western and central Alps, while it is the lower plate of the thrust systems in the Apennines and the Dinarides (e.g. Schmid et al., 2008; Handy et al., 2015).

- Q: Section 2.2.2: trivial, suggestion to rather include typically number of used stations over magnitudes

  *A: We agree with the reviewer that the amount of knowledge gained from section 2.2.2 on the relation between distance and magnitude is limited. We decided to remove the section from the manuscript. Instead, we provide the most important information in the introduction to the methodological tests. We hope that this is fine with all reviewers.*
  *We had a look into the statistics of stations used per magnitude unit. While it is possible to define some general upper and lower bounds, the total number of stations which can be used is influenced by multiple factors, such as the location of a station within the network (at margin or in middle; how close to denser Swath-D or other sub arrays). Additionally, the number of available stations in the AASN varies over time. Furthermore, SNRs depend on daytime, season and location. Reporting on all these factors would lengthen the manuscript. Therefore we decided not to include more detailed statistics on the used number of stations over magnitude.*

  *→ p. 9 in the annotated manuscript:*

  We benefit from the large seismic network and use more than 80 stations at distances of up to 400 km for the largest events (Fig. 3). For earthquakes with moderate magnitudes between Mw 3.5-3.9 we mostly rely on 20 to 50 stations within a radius of 200 km. The number of available stations depends on the magnitude, but also on the epicenter location within the network. Furthermore, the SNR and quality of the individual stations is variable in time and space. Before the inversions, we applied the toolbox *AutoStatsQ* to identify seismic stations with misorientations, metadata errors or gain problems (Petersen et al., 2019).

  **removed**:

  ## 2.2.2 Magnitude-distance relation
  The distance range in which stations can be used for MT inversion strongly depends on the event magnitude. While we use an epicentral radius of less than 100 km for the smallest events with magnitudes Mw 3.1-3.3, epicentral distances may be as large as 300-400~km for the largest events with magnitudes greater Mw 4.0. For events between Mw 3.5-3.7 and Mw 3.8-3.9, we use distances of up to 160~km and 200~km, respectively. This results in different inversion set-ups: the number of available stations varies between less than 10 stations for the Mw 3 events to above 80 stations for the largest events. Fig. 6 illustrates this relation. The left panel shows waveforms of an Mw 4.1 event in Switzerland at distances of up to 350~km. Even though the SNR decreases with distance, a distinct Rayleigh wave can be seen. The second event from 2017-10-27, France, has a magnitude of Mw 3.6. For distances larger than 160km, SNR are very low for most stations. We did not remove stations with generally high noise levels from the plot to illustrate that a careful rejection of very noisy and disfunctional stations is required. We apply the toolbox *AutoStatsQ* in advance to identify seismic stations that are misoriented, have errors in their metadata or gain problems (Petersen et al., 2019).

- Q: p. 13, Fig. 5d . strike 1 and 2 directions: "too carnival to color them with (b) colors for each kind of solution?"

  *A: We tried to plot the strike directions colored by mechanism type, as suggested, but the result was not satisfactorily. On one side, colors were hard to identify on the thin lines and on the other hand, increasing the line widths result in overloading the figures. Note that, for our discussion, it is not needed to know which line corresponds to which mechanisms, but more to get an idea of the variability among the solutions. Therefore, we finally decided to keep the gray lines.*

- Q: p. 22: 75 EQs over how many occurring?

*A: We did not provide the total number of earthquakes here, because the number strongly depends on the lower magnitude threshold. The local magnitude that is reported by the different catalogs may vary by 0.1 to 0.3 units, resulting in very different estimates of the ratio of events we could invert for. Across the Alps, we successfully inverted about 80 % of events with Ml > 3.3 compared to the GEOFON catalog. This is described by the term "most earthquakes". In order to show that we have significant problems with the smaller events, we state that we only obtained stable solutions for one third of events with Ml 3.1 to 3.3 reported in the GEOFON catalog.*
*We do not think that it would be helpful to give additional numbers for different catalogs here, as it might be very confusing to compare the different local magnitudes first in order to get a better estimate of successful inversions in dependence on the magnitude range.*

*→ p. 19 in the annotated manuscript:*

We obtained deviatoric MT solutions for 75 earthquakes occurring between 01/2016 and 12/2019 in the wider Alpine region, for which we determine moment magnitudes between Mw~3.1 to 4.8 (Fig. 9, Supplement Table S1). While we were able to compute stable MTs for most Alpine earthquakes from regional catalogs with local magnitudes larger Ml 3.3, we resolved only thirteen MTs for earthquakes with local magnitudes between Ml 3.1 and 3.3, corresponding to one third of the events in this magnitude range compared to the GEOFON catalog. Low SNR in the tested frequency bands covering frequencies between 0.02 and 0.5 Hz and less available stations hindered successful inversions for the other small earthquakes. Furthermore, we realized that a station spacing of about 60~km is not sufficient for small earthquakes (Mw<3.3) in case a part of the data is rejected due to quality issues.

- Q: Describe merging of MT bulletins:
  *A: We added the information on merging the MT bulletins on p. 21:*

  Whenever more than one MT solution is available from the different bulletins, we prioritize local institutes (INGV for Italian earthquakes, SED for earthquakes in Switzerland, ARSO for Slovenia), unless they indicate high uncertainties. Furthermore, EM-RCMT with great experience for the Mediterranean and surrounding areas is favored over GEOFON solutions and over GCMT.

- Q: Describe colors in Fig 9

  *A: We follow the suggestions of the reviewer and explain in the subscript of Fig. 9 the colors of the focal mechanisms. As the clustering is based on a clustering procedure which relies on the smallest rotation between the mechanisms, we added a small description to link to the text in which the details for the clustering approach are provided.*

  Figure 9. Moment tensor inversion results from 01/2016 to 12/2019 (focal spheres with black lines) along with MTs from 1983-2015 from bulletins of GCMT, GEOFON, INGV, SED, EM-RCMT and ARSO (lighter colors). Similar colors represent clusters of comparable mechanisms, obtained from a clustering approach based on the smallest rotation between the mechanisms (see text). Red and orange colors correspond to dominant normal faulting mechanisms in a cluster. Thrust faulting is indicated in blue and strike-slip faulting earthquakes are colored in green and purple. [...]

- Q: All place names to maps

  *A: Thanks for pointing out that the introduction and the discussion is much easier to understand when adding the names of places, faults and countries to the map. We added them in the overview map in the introduction. Furthermore, we included convergence rates and improved the readability of the figure.*

- Q: p. 20 – redundant information in sec 4.1 with event section:
  *A: We followed the suggestions by reviewer 2 and 3 and restructured the results and discussion. In doing so, we removed redundant information.*

- Q: Comparison to world or italian stress map

  *A: We follow the suggestion of the reviewer and add short comparisons to the European stress map (Heidbach et al. 2016) in the discussion Sec. 4.1 (p.23).*

  → *p. 23 in the annotated manuscript:*

  At the southern margin of the central Southern Alps, we observe predominantly thrust mechanisms with NNW-SSE to NW-SE oriented P-axes in the central Alps, close to Lake Garda, to NNE-SSW oriented P-axes further east, close to Vicenza (Fig. 11a, features d and e). Our stress inversion results confirm dominating compression from central to the eastern the central to eastern Southern Alps with sub-horizontal σ₁ orientation (Supplement Fig. S2), which is in agreement with the stress map of the Mediterranean and Central Europe (Heidbach et al., 2016). Seismic activity at thrust faults originating from the N-S convergence of the Adriatic and Eurasian plates in the Southern Alps are well known and have been described by various studies (e.g. Pondrelli et al., 2006; Anselmi et al., 2011; Poli and Zanferrari, 2018). According to Cheloni et al. (2014), the SE Alpine thrust front absorbs about 70% of the convergence between the continental plates. In the transition from the SE Southern Alps to the Northern Dinarides, across a distance of 200 km, a rotation of the P-axes from NW-SE in the western part to NNE-SSW in the eastern part is observed (Fig. 11a, features a-c). Despite increased uncertainties due to the relatively low number of available MT solutions, we observe a similar the same rotation of σ₁ . Although less distinct, this rotation can also be seen when looking at the stress direction obtained from thrust MTs in Heidbach et al. (2016). The changes in the orientation of the thrust mechanisms may be attributed to the bending of the southern thrust front of the Alps and to the transition to the strike-slip fault systems in the Dinarides.

  Heidbach, O., Custodio, C., Kingdon, A., Mariucci, M. T., Montone, P., Müller, B., Pierdominici, S., Rajabi, M., Reinecker, J., Reiter, K., Tingay, M., Williams, J., and Ziegler, M.: Stress map of the Mediterranean and Central Europe 2016, GFZ Data Services, https://doi.org/10.5880/WSM.Europe2016, 2016.

- Q: p. 22: The manuscript you compare with is not still published but under review (with also a major revision asked). So It is not (still) reasonable to confirm the results obtain in this study.

  *A: We are thankful for pointing out the ongoing discussion on this paper and removed the comparison. We also excluded Mader et al, which is still in review. If it is published before our paper we might include it again.*

- Q: p. 22, l. 434: Only here you can finally use the term "stress". Before this analysis you were always describing only deformation (see previous comments).
  *A: Thanks for pointing out this mistake. We agree and now avoid saying stress regime in the paragraphs before we actually write about the stress inversion results.*

- Q: p. 24: Close to historical events of Basel, another small cluster (green) is mapped in Fig. 12a, but you do not describe it. Why? low energetic?

*A: The green cluster is not related to the historical event of Basel, which was >50 km farther to the North. We concentrated in our analysis on the larger clusters for which we were able to obtain multiple moment tensor solutions for earthquakes between 2016 and 2019. However, we agree that we do not sufficiently explain this in the text and added this information now. For the green cluster at the Swiss-Italian border, we have only two solutions directly within the cluster.*

*p. 27 in the annotated manuscript:*

We focus on the analysis of the largest clusters, for which we were able to obtain multiple moment tensor solutions between 2016 and 2019.

- Q: p. 24: which is the amount of information given by your focal mechanisms? without them you would get the same results or not?

  *A: We did not sufficiently link our CMT solutions and the seismicity and strain section. We restructured the entire discussion and included more comparisons to literature. Now this part is used to discuss the CMT solutions in their seismological and tectonic context, allowing us to characterize the different faulting style-domains in the Alps.*

- Q: p. 25: Cumulative seismic moment map → Unit? Nm? I feel some white dots are for grid elements where zero to quasi zero seismicity is present. Sure to avoid to map points where cumulative M0 is zero?
  A: *Thanks for pointing out that we missed the unit [Nm], which is now added to the plot . We checked again, points where the cumulative M0 is zero are not plotted.*

- Q: p. 26: Seismic moment: So it is really significant to add this data to this work? this last paragraph seems to scarcely contribute to the discussion.

  *A: We hope that now, with the restructured discussion, it is more clear that the spatial distribution of the cumulative seismic moment is an interesting feature to compare to the strain and uplift rates and to the faulting styles. It shows that seismicity is significantly stronger in the SE Alps, where thrust faulting related to the convergence of Adriatic and European plate is accompanied.*

- Q: p. 26: this paragraph and this part of the study may benefit with a comparison with Sernai et al. 2019, Earth Sceince Review and reference therein

  A: *We are thankful for the suggestions of reviewer 1 and 2 to include comparisons with Sternai et al. 2019, Delacou et al. (2004) and others. We think that the discussion improved a lot considering these studies on the seismotectonic setting.*
  *As these changes and new comparisons affect the entire last part of the discussion, we refer to the annotated manuscript instead of showing all modified text blocks here.*

In this paper, the authors provide MT inversions homogeneously performed on the whole European Alpine belt thanks to the Alp Array initiative and corresponding dense seismic networks (AASN). They rely on the seismicity recorded during 4 years at ~600 stations to derive 75 moment tensors of associated local magnitude ranging between 3.1 and 4.8. In the first part of their manuscript, they present a set of methodological tests in order to define the best input datatypes and the best parameters to constrain their moment tensor inversions. These methodological tests moreover allow them to establish moment tensor inversion guidelines for areas of similar tectonic context and of small to moderate seismicity. The second part of their manuscript very briefly describes the 75 well constrained moment tensors retrieved from applying the aforementioned guidelines to the Alpine/Apenninic areas over the 2016-2019 AA seismic data. The last part of the manuscript deals with the representativity of the patterns they identify in the seismicity (focal depths, seismic clusters) and in the tectonic regimes of the Alpine belt and surrounding areas with regards to seismotectonic, geodetic and historical contexts.

The paper is well written, and mostly well organized, which makes it fluent and pleasant to read, even if the different parts could be better balanced (see below and following specific comments). I highly appreciated the methodological testing section, which delivers several conclusions which will certainly be of interest to a broad community of both seismologists and tectonicians (minimum station coverage required depending on geological clues, min and max epicentral distances, contribution of tensile faulting). They indeed apply a very thorough and rigorous analysis of the potential biases associated to either the dataset, the network or the inversion parameters. These tests allow the authors to estimate the % of non-DC components in their MTs, as well as their possible meaning, and to assess the resolution of the DC components. The very first part of this section could be condensed though (see specific comments below).
Section 3 presents some very interesting results, that the reader may wish to see deepened and better highlighted.

The discussion part (section 4) however mixes up some aspects which in my view represent new computations and result descriptions (families identification through clustering, depth distribution) with analysis of the results and comparisons with other studies, which makes the take-home message difficult to grab. I suggest reorganizing sections 3 and 4 as specified below. Most importantly, the results lack of interpretation and would benefit from being put into their broader tectonic and geodynamic contexts. An attempt to do so is sketched in section 4 but this should be extended (i.e. present-day plate kinematics and limits, crustal units, Moho depths...see specific comments). Similarly the discussion would benefit from a more detailed comparison of the main findings with previous studies, and lacks of references to recent Alpine large scale studies.

Lastly, I recommend rewriting the conclusions to better highlight the major findings of the study and main contributions brought to the seismological and seismotectonic communities, as well as the potential applicability to other tectonic domains/study areas.

I leave to the editor to appreciate whether the paper requires minor or moderate revision based on the following specific comments. In any case, I strongly believe that this paper will make a valuable contribution to the Alpine -and possibly broader- community after a few improvements.

*A: We followed the detailed suggestions by reviewer 3 to restructure the results and discussion sections, to include more comparisons to previous studies and to rewrite the conclusions. We feel that these modifications improved the manuscript a lot.*

**Specific comments:**

- Q: The introduction is well written and exhaustive. Referring to Figure 1 sooner (§3) would help the reader when describing the various predominant tectonic regimes throughout the

belt though.

A: *We follow this suggestion and refer to Fig. 1 earlier now and move the figure up. Additionally, we followed the suggestions by reviewer 2 and added more information on faults and places that are mentioned in the text.*

- Q: I would suggest making subsection 2.1 more synthetic. For example, keep in pages 7 and 8 only the details about the choices you made, especially since you provide detailed comparisons of input data types and various combinations in section 2.3.

  A: *We discussed among the co-authors and agreed that we would like to keep the descriptions of the different data types that can be used in the inversion (previously p. 7 and 8) in the beginning of the methodological section. We think it is helpful to present the possibilities of the MT inversion tool before showing the different tests which we perform. In section 2.3 we show a test on these input data types, but we do not repeat the descriptions of these. The input data types also need to be explained in advance for the other tests so that we can justify which choices we make. We assume that this is also in agreement with reviewer 1 and 2, who suggested only minor changes in the methodological part of the manuscript. If the reviewers and editors not not agree with this structure, we are of course willing to follow the suggestion by reviewer 3.*

- Q: section 2.2 presents some nice methodological conclusions, that should be emphasized as stated l.208: why not summarize these guidelines at the end of the conclusions of the paper, in order to open the discussion towards applicability to other similar contexts ? For example stating that for the specific Alpine context or for other densely instrumented low seismicity areas, DC component is well resolved whether allowing for a CLVD or isotropic component or not as mentioned l.236., or that a small number of stations/small azimuthal coverage may be sufficient depending on their orientation wrt to the strike of the fault as stated section 2.2.4, or that CLVD and isotropic components cannot be distinguished reliably.

  A: *We agree with the reviewer and followed all of these suggestions. We rewrote the conclusions accordingly to highlight the results of the methodological tests.*

- Q: Section 2.2.1: paragraphs between l.226-252 would better fit before Figure 3 and its corresponding §, since they give important explanations to understand the tests that are implemented, and since the last § of page 11 is the direct continuation of the first two § of this subsection.

  A: *We followed this suggestion and moved the according paragraph upwards before Fig. 3 and its paragraph.*

- Q: l.298: what are the 8 combinations mentioned? Only 6 are presented in Figure 7. Are the other two envelopes-td and envelopes-cc combinations? This is confusing. As the latter two are only presented in the supplement I would not mention those here.

  A: *This information is indeed confusing. The Number eight refers to tests in the main text and the supplement. We removed the number and instead add the reference to the supplement here.*
  → *p. 16 in the annotated manuscript:*

We perform MT inversions using  different combinations of input data types (Fig. 7 and Supplement Fig. 1): [...]

- Q: l.306: «low uncertainties»: how much? It could help the reader to add the uncertainty color scale to Figure 7 at least.
  A: *We followed the suggestion by the reviewer and added a color scale to Fig. 7 showing the mean standard deviation of six MT components.*

- Q: l.401 : « draw a more detailed picture of the seismic activity in this study area » : what are these new detailed features compared to the literature in the end? The new characteristics that could be derived thanks to this study should be emphasized in the conclusions, rather than summarized as was already done in the abstract.
  A: *We followed the suggestions and rewrote the conclusions. We now emphasize better the findings of both the methodological tests and the characteristics of the resulting MT solutions in the seismo-tectonic domains.*

- Q: sections 4.1 and 4.2 are merely discussion sections since new results are presented here from applying various clustering algorithms, and especially since no interpretation/analysis is done of the described features concerning tectonic regimes and depth distributions. No explanation is provided for the similarities or discrepancies that are highlighted between the different areas. I don't find any comparison to previous study here either. I would suggest moving these subsections into the result section 3.

  *A: We are thankful for the comments on the structure of results and discussion. We follow the suggestion by reviewers 2 and 3 and restructured them. The clustering of focal mechanisms of previous section 4.1 and the entire chapter 4.2 were moved to the Result section. We keep the discussion on stress and P/T axis in the Discussion and enhance their discussion in the seismo-tectonic context including comparisons to existing seismological and tectonic studies.*

- Q: Figure 10 would be clearer if grouping stereograms by family...maybe try plotting a single stereo for each family identified through the clustering algorithm ?

  *A: We plotted the distribution of the P and T axes for each family separately (see below). However, we discussed among the co-authors and came to the conclusion that for us the joint figure with the different colors per family is more comprehensive. If it is however wanted by the reviewer and the editor, we can easily exchange the figure.*

[Figure]

*Figure 1: Alternative version of Fig. 10 showing the P and T axis per faulting type separately.*

- Q: You describe a rotation of the P-axes in the SE Alps, is it a new feature or was it already observed by previous studies? If so, do you provide more constraints/increased resolutions on the corresponding orientations? How was it interpreted?

  *A: We added a paragraph to the discussion section with the information, that such a rotation can also be seen in the European stress map and may be related to the bending of the southern most thrust front.*

  *→ p. 25 in the annotated manuscript:*

  At the southern margin of the central Southern Alps, we observe predominantly thrust mechanisms with NNW-SSE to NW-SE oriented P-axes in the central Alps, close to Lake Garda, to NNE-SSW oriented P-axes further east, close to Vicenza (Fig. 11a, features d and e). Our stress inversion results confirm dominating compression from central to the eastern the central to eastern Southern

Alps with sub-horizontal σ $_1$ orientation (Supplement Fig. S2), which is in agreement with the stress map of the Mediterranean and Central Europe (Heidbach et al., 2016). Seismic activity at thrust faults originating from the N-S convergence of the Adriatic and Eurasian plates in the Southern Alps are well known and have been described by various studies (e.g. Pondrelli et al., 2006; Anselmi et al., 2011; Poli and Zanferrari, 2018). According to Cheloni et al. (2014), the SE Alpine thrust front absorbs about 70% of the convergence between the continental plates. In the transition from the  Southern Alps to the Northern Dinarides,  a rotation of the P-axes from NW-SE  to NNE-SSW  is observed (Fig. 11a, features a-c). Despite increased uncertainties due to the relatively low number of available MT solutions, we observe a similar  rotation of σ $_1$. Although less distinct, this rotation can also be seen when looking at the stress direction obtained from thrust MTs in Heidbach et al. (2016). The changes in the orientation of the thrust mechanisms may be attributed to the bending of the southern thrust front of the Alps and to the transition to the strike-slip fault systems in the Dinarides.

- Q: Same question for the transition from thrusting to strike-slip in the SE Alps-Dinarides junction. I guess there is an attempt to do so afterwards l.481-484 but I'm not sure whether it refers to the same feature.

  *A: The Northern Dinarides at the transition to the Alps have been studied by Moulin et al. (2016) and Pondrelli et al. (2006). The area is known for right-lateral strike-slip systems. We added this information in the manuscript:*

  *→ p. 25 in the annotated manuscript:*

  The transition from dominant thrust faulting  close to Friuli to the strike-slip events to the east and in the northern Dinarides was also described by Pondrelli et al. (2006) and is mapped by the change from a sub-vertical to an almost horizontal σ $_3$ direction (Supplement Fig. S2). Moulin et al. (2016) describe right-lateral motion ($3.8 \pm 0.6$ mmyr$^{-1}$) on three main Dinaric faults, and suggest that the system of NW-SE oriented right-lateral strike-slip faults might be the north-eastern boundary of the Adriatic microplate.

- Q: l.434-438 should definitely appear in section 3 rather than here.
  A: *We restructured results and discussion. We moved all parts related to the clustering of focal mechanisms to the results, but we keep these few lines referring to the stress inversion in the introductory part of the discussion on the deformation and stress regimes. However, we make sure to discuss our results in the context of literature on tectonics and compare to the European stress map. We do provide the methodological details in the supplement.*

- Q: If the authors wish to keep section 4.2 in the discussion, they should give more elements to discuss the observed depth distribution : how are the observed depth variations explained by the various tectonic contexts ? How does the depth distribution of the events correlate with lateral depth variations of the different Mohos ? In the seismotectonic context part of the introduction the hypothesis of subduction polarity reversals at the transition with Apenninic and Dinarid slabs is mentioned. Are centroid depths deeper in these places, shallower, or not systematically different from the surroundings ?

  *We followed the suggestion by the reviewer and moved this section to the results. Additionally, we included a small paragraph in the new discussion part 4.2 indicating, that we do not find lateral depth variations which can be attributed to different Moho*

*depths. We can not interpret any pattern related to a polarity switch in the Eastern Alps as proposed by Lippitsch et al. 2013. On one hand, the existence of this switch is still under debate, and on the other hand, the dominantly shallow events do not show systematic changes in depth in the surrounding. Beneath the Northern Dinarides, there is no (more) slab.*

→ *p. 30 in the annotated manuscript:*

The observed depth ranges of our MT solutions (Fig. 10) are in accordance with the maximum depths in the long-term seismic catalogs of gGCMT, INGV and GEOFON, including >50,000 earthquakes with Ml>2.0 (Fig. 12c). While the Moho depth increases gradually from less than 30 km at the northern margin of the Alps to above 50 km in the central part of the orogen (Spada et al., 2013), we do not observe any gradual change in the event depth. These catalogs and our own centroid depths show that seismicity is shallow across most of the Alps with rare deeper events (<30 km) at the southern margin, where the Moho is at about 40 km depth (Spada et al., 2013). These few deeper events are located above the Moho., while mMaximum depths of above 60 km are observed in the Apennines.

- Q: In the caption of Figure 11 is written : « the outlines of spatial clusters of increased seismic activity from Fig12a are indicated for comparison ». Why ? Where is this comparison made in the main text ? Should we expect any correlation between higher seismicity rate clusters and depth variations ? If yes it would be valuable to give a more detailed comparison/explanation. If not, why not rather plot the outlines of the different tectonic regime families identified in Figure 9 and section 4.1 ? And, if possible, it would be helpful to also outline the different plate limits on these maps.

  A: *We followed the suggestion and added the main faults and deformation fronts on the maps showing the centroid depths.*
  *We indicate the outlines of the seismicity clusters of Fig. 12 in Fig. 10 as a way to allow an easier orientation on the map. So that is possible to see the centroid depths of events within each seismicity cluster. We do not expect a correlation of depth and seismicity rate in general, although we observe that within the Alps the deepest events are at the south-eastern margin where also seismicity is highest.*

  *We modified the caption of Fig. 10:*

  The outlines of spatial clusters of increased seismic activity from Fig. 12a are indicated for comparison orientation.

  *In Fig. 10 we show the centroid depth as colored focal mechanisms and sort them by mechanism type. Therefore we feel that adding the domains from faulting style/tectonics as observed in Fig. 9 would show redundant information. Furthermore, the outlines of the mechanism families are less distinct compared to the seismicity clusters. In areas like the SE Alps, the N Dinarides and Lake Garda, these domains are well described by the seismicity clusters which we show. In the W Alps, the mechanisms are more heterogeneous. Furthermore, we now added representative MT solutions (where applicable) to the cluster in Fig. 12 to better link the MT solutions and the seismicity clusters.*

- Q: section 4.3: in my view, the discussion part really starts here, but should be extended: How do the higher seismicity rate areas relate to the tectonic regime families formerly identified? This is addressed briefly at the end of p.24 but it would be interesting to cover it in more detail. For example Figure 12a could benefit from adding a representative focal mechanism for each cluster. Or maybe overlay the stress tensors/stereos from the supplement. This would help the reader to get a more general picture of the regional seismicity and to put back the results into a broader context. This would support and extend the discussion of the different features made l.481-493. It would also be helpful to refer to the colors of the clusters Fig 12a) when analysing them in the main text (or at least display numbers from Fig1 on Fig12a). It would be easier for the reader as well to structure the discussion by bullet points for each identified family/feature (or by small paragraphs with bold header as was done in the methodology section 2.1).

  *A: We followed the suggestions by reviewer 3 and now relate our MT inversion results more clearly to the seismotectonic context.*
  *In order to do so, we added representative mechanisms in Fig. 12a and describe them in a the paragraph on the seismicity clusters. Furthermore, we now included in Fig. 12a the same numbering used in Fig. 1 and refer to these numbers in the entire discussion. We agree that this makes it much simpler to follow the description.*
  *Whenever necessary, we exchanged the names of clusters or regions to make it more clear about which regions we are talking.*
  *We partly follow the last suggestion to structure the discussion by region. We do not use bullet points but reordered the paragraphs in the first part related to the mechanisms, P and T axis and stress inversion. We now explain in advance that we focus first on the "typical" compressional regimes at the south-central to SE margin of the Alps, before we discuss its transition to the strike-slip regime in the N Dinarides and the extensional regime in the W Alps.*

  *In the second part of the discussion, we wish to keep the overview over the entire Alps, with a focus on the comparison of the different features shown in Fig. 12. We rely on the numbering of the seismicity clusters/regions and on coherent naming of regions, but keep the discussion in the order of the features shown in Fig. 12.*
  *In both parts of the discussions we added new comparisons to literature on seismic activity and tectonics of the Alps and discuss our results in the geological and tectonic setting.*

  *Because of the large number of reordered paragraphs and additionally added information, we do not show the discussion here but refer again to the annotated manuscript.*

- Q: The discussion would also benefit from analysing the repartition of seismicity and faulting styles in the light of the complex tectonic context mentioned in the introduction. These results could be discussed with regards to current plate kinematics (spatial variations

of convergence rates, counterclockwise Adria rotation...). For example indicate current plate limits and kinematics with arrows on the maps of Fig 12.

*A: As mentioned in the answer to the previous question, we rewrote parts of the discussion and included more comparisons to the tectonic features mentioned in the introduction as well as recent GNSS studies. We included the thrust fronts (which act as the boundaries of the Adriatic plate) and the most important faults in Fig. 12b-f. We now includes arrows with convergence rates on the map in Fig. 1 and discuss the results also with respect to plate kinematics. Additionally, we show in the supplement recent horizontal GPS velocity measurements.*

- Q: I really appreciate the effort to compare seismicity with representative measurements of geodetic deformation (i.e. spatial gradient of uplift rate as a proxy for vertical strain). For the map of the maximum shear rate, are the results similar to those which would be obtained using the 2nd invariant of the strain tensor?
  *A: In the preparation of the manuscript we plotted both the $2^{nd}$ invariant of the strain tensor and max. shear strain. Both plots show very similar patterns. We had already modified Fig. 12f to show the $2^{nd}$ invariant of the strain tensor instead of the maximum shear strain rate before the original submission, because it is a more common approach to show horizontal strain rates. Unfortunately we forgot to remove the old statement in the caption and supplement, which falsely said that we show the maximum shear strain rate. We corrected the caption of the figure and the supplement accordingly.*

- Q: Beachballs on Figures e) and f) are overloading the global picture. As mentioned above, I would rather display a representative mechanism, or stereo, or stress arrow on Figure 12a, and leave the sole seismicity distribution here (or even better, seismicity density, as isolines maybe), especially if the authors wish to focus the discussion on the areas presenting both higher seismic rates and higher geodetic deformation.

  *We added representative MT solutions to Fig 12(a) and removed the mechanisms from (e) and (f). To facilitate the comparison to MT solutions, we still use the color-coded dots for events with MT solutions. This allows a comparison of MT solutions and uplift gradient/ horizontal strain rates also in areas which cannot be represented by the single representative MT solutions per cluster.*

- Q: l.505 : I disagree on the relation between vertical geodetic gradients and seismic activity. As stated by the authors themselves in the following paragraph, several areas show a spatial decorrelation between higher vertical gradients and seismicity (SW Alps, E Po plain).

  *A: We did not formulate this well. We added few words to make sure that the vertical gradients and seismic activity do not show the same pattern across the entire study area. As pointed out by the reviewer we state later that e.g. in the E Po plane the spatial gradient of uplift rates is high while the seismicity is low. The high vertical gradient results here from the subsidence of the Po plane relative to the surrounding.*

***On p. 30 in the annotated manuscript we now state:***

Within the Alpine mountain range, the GNSS data shows a consistent uplift relative to the surrounding areas (Supplement Fig. S3). Fig. 12e and f emphasize the relation between recent seismic activity and both, high spatial gradients of the uplift rate (e) and the shear strain rate (f) across large parts of the study area.

[...***Details are then provided in the paragraphs below.]***

- Q: l.520 : representing the absolute vertical gradient indeed probably biases your comparison between tectonic uplift and seismicity occurrence. Why not instead represent positive vertical gradients only ?

   *A: On Fig 12a we plot the absolute value of the spatial derivative of the uplift rate. If we would not take the absolute value and instead show the vertical gradient only, the resulting pattern would strongly depend on the direction of computing the spatial derivative. E.g. looking from NW to SE would result in a totally different picture than looking from SE to NW and so on. When having in mind a profile instead of a map, this would work. Also, if the Alps would not be bending in the Western part, we could easily define the spatial derivative relative to the ridge of the Alps. However, including the bending and also the neighboring Apennines and Dinarides we could not think of a method to show only the positive value of a spatial derivative.*

- Q: On the contrary the correlation between cumulative seismic moment (12d) and shear strain rates is striking and should be emphasized.

   *A: Thanks for pointing out this, we added a statement to the according paragraph.*

   → *p. 30 in the annotated manuscript:*

   The distribution of the cumulative seismic moment (Fig. 12d) agrees particularly well with the distribution of shear strain rates (Fig. 12f).

- Q: This comparison is really interesting but lacks references. How are these correlations supported by other studies (regional e.g. Serpellonni et al., 2016 https://doi.org/10.1016/j.tecto.2016.09.026, or local, e.g. Anderlini et al., 2020 https://doi.org/10.5194/se-11-1681-2020 ?

   *A: We are thankful for pointing out that we did not compare our results to recent GNSS studies. As mentioned in the answers to the previous questions related to the discussion we now included comparisons to the mentioned studies and some others, which clearly helps to discuss the observed patterns in the context of tectonic movements.*

- Q: The conclusions do not present the same quality (substance and form) as the rest of the paper. They should be rewritten to participate in highlighting this fruitful study. The major findings of the paper should be more clearly emphasized, including the ones concerning the seismotectonic insights. For example it is a major finding in my opinion, which will moreover be of interest to a broad community, to note that with 4 years of acquisition of

small to moderate earthquakes, the authors are able to derive seismotectonic domains which are representative in faulting style of those derived from longer term seismicity/higher magnitude events ! Maybe state it more clearly ~ l.540. The very last § of the conclusions is not appropriate here. It should appear at the end of the methodological section instead. I suggest an opening focused on the applicability of the methods to other study areas and summarizing the main guidelines for similar tectonic context in order to be able to derive reliable and representative MTs over short time span and dense networks (see suggestions in the above comment related to section 2.2).

*A: We are thankful for these helpful comments and suggestions on the conclusions. We rewrote large parts of the conclusions to emphasize the findings of the paper with respect to seismotectonic insights (and, as previously asked for, with respect to the methodological tests).*

**Technical issues, English and typos:**

- l.30 and in several other places : « both » not « both, »

*corrected.*

- if possible uniformize English or American English spelling (characterised vs catalog for example)

*Thanks for the hint. We proof-read again and tried to use American English only.*

- l.81 « data points » : polarities ? Stations ?
*Corrected.*

- Figure 1 caption : datasets) .

*Corrected.*

-l.88 and everywhere at the beginning of sentences : « Furthermore » not « Further »

*Corrected.*

-l.119 « an »

*Corrected.*

-l.126 « *Grond* »

*Corrected.*

-l.130 : « firstly » [ …]. Secondly, […] »

*Corrected.*

-l.133 « Grond »

*Corrected.*

-l.134 « bootstrap (BS) chain »

*Corrected.*

-Figure 2 caption : « are indicated above each column ».

*Corrected.*

- Figure 2e) : mxx instead mnn and so on ?

> *A: We use routinely the East, North, Down convention in the MT inversion, therefore we keep the MT components mnn, mne, ...*

-l.183 « sensitive to »

*Corrected.*

-l.212 representative … in terms of what ? Magnitude range ?

*Information was added.*

-l.240 « If[…], it is however […]. »

*Corrected.*

-l.269 and in several other places « in the case of », not « in case of » + « without any station covering »
*Corrected.*

-l.275 : « one must assess the »
*Corrected.*

-l.322 : Fig. 7)

*Corrected.*

-l.367: a limited a azimuthal

*Corrected.*

-l.369 « any *a priori* »

*Corrected.*

-Figure 8 caption : « radius r given below each column ». Refer to section 2.1 and Fig 2 for the fuzzy Mts.

*Corrected.*

- Figure 9 caption : I would have expected more solutions from the 1983-2015 bulletins over the whole area : is it due to the minimum Ml 3 threshold ? Or are some MT solutions missing from example from Geoazur ?

> A: *We did not find more moment tensor solutions provided online. We checked all catalogs. From Geoazur we found only solutions starting in 2019.*
> *There are many first motion based focal mechanisms available in various paper. However, for the reasons mentioned in the introduction, we do not use these mechanisms here but concentrate on the analysis of full-waveform based moment tensor solutions.*

-l.381 : « (Fig.9, Supplement Table X).

*Corrected.*

-l.393: « section 4.2 »

*Corrected.*

-l.421-422 : the two sentences are redundant.

*Corrected.*

-Figure 10 : add a), b), and c) labels on the subfigures.

*Corrected.*

-l.426 : no stresses here, but regimes or horizontal T-axes

*Corrected.*

-l.454 : « at shallower depths »

*Corrected.*

-l.461 and whenever referring to historical seismicity/events : « historical » not « historic »

*Corrected.*

-l.488 : « mechanisms. In addition »

*Corrected.*